# Mining the Species Diversity of Lacewings: New Species of the Pleasing Lacewing Genus *Dilar* Rambur, 1838 (Neuroptera, Dilaridae) from the Oriental Region [note 1]

**DOI:** 10.3390/insects12050451

**Published:** 2021-05-14

**Authors:** Di Li, Horst Aspöck, Ulrike Aspöck, Xingyue Liu

**Affiliations:** 1Department of Entomology, China Agricultural University, Beijing 100193, China; ld_77c@cau.edu.cn; 2Institute of Specific Prophylaxis and Tropical Medicine, Medical Parasitology, Medical University of Vienna, Kinderspitalgasse 15, A-1090 Vienna, Austria; horst.aspoeck@meduniwien.ac.at; 3Zweite Zoologische Abteilung, Naturhistorisches Museum Wien, Burgring 7, A-1010 Vienna, Austria; ulrike.aspoeck@NHM-WIEN.AC.AT; 4Department of Integrative Zoology, University of Vienna, Althanstraße 14, A-1090 Vienna, Austria

**Keywords:** Dilaridae, *Dilar*, new species, species diversity, distribution, Oriental region

## Abstract

**Simple Summary:**

The pleasing lacewing (Dilaridae) is a little known family of the holometabolous order Neuroptera, and our understanding of their species diversity has long remained poor. Here, we present descriptions of 12 new species of the pleasing lacewing genus *Dilar* Rambur, which is widely distributed in the Palaearctic and Oriental regions. We found disparate wing marking patterns as well as several unique characters of the male genitalia of the new species, which highlight the diverse morphologies of *Dilar*. Based on a faunal analysis, eight areas of endemism of *Dilar* were distinguished, and the state of their species diversity and endemism were summarized. The Oriental part of China was revealed as the region with the highest species diversity of this genus, and Yunnan within this region stood out as the most species-rich subregion.

**Abstract:**

The species diversity of insects is extraordinarily rich, but still has been insufficiently explored or underestimated particularly for uncommon groups. The pleasing lacewings (Dilaridae) are a little known family of Neuroptera with distinct sexually dimorphic antennae. The species diversity of pleasing lacewings was recently found to be severely underestimated and requires a comprehensive investigation, as well as systematic reviews. Here, we report on 12 new species of the pleasing lacewing genus *Dilar* Rambur, 1838, from the Oriental region, namely *D. forcipatus* sp. nov. and *D. laoticus* sp. nov. from Laos (new country record of *Dilar*); *D. malickyi* sp. nov., *D. phraenus* sp. nov. and *D. rauschorum* sp. nov. from northern Thailand; *D. striatus* sp. nov. from northern Vietnam; *D. cangyuanensis* sp. nov., *D. daweishanensis* sp. nov., *D. nujianganus* sp. nov., *D. weibaoshanensis* sp. nov., *D. yucheni* sp. nov., and *D. zhangweiae* sp. nov. from Yunnan and Tibet, both in southwestern China. The new species of *Dilar* display several types of wing marking patterns, and the morphology of the male genitalia is highly diverse. A comprehensive examination of the species diversity and distribution of *Dilar* concluded that Yunnan (southwestern China) represents a biogeographic region with high endemism and the richest species diversity. The potential correlation between vertical distribution and geographical latitude in *Dilar* was also analyzed.

## 1. Introduction

The global decline of insect species and their abundance is becoming drastic due to climate change and human-derived perturbations [1]. Assessment of such insect defaunation is urgently required but requires standardized, quantitative methods based on rich data from long-term monitoring [2]. Currently, exemplar studies on assessment of insect biodiversity are mainly performed for certain well-studied taxa such as butterflies and bees, and many works are regionally confined to developed countries from North America and Europe [2,3]. However, basic data on the species diversity of uncommon insects from developing or undeveloped countries, which is vital for an accurate assessment of global insect biodiversity, are seriously lacking due to insufficient sampling and taxonomy. The superorder Neuropterida, comprising Raphidioptera (snakeflies), Megaloptera (dobsonflies, fishflies, alderflies) and Neuroptera (lacewings), is an archaic group within Holometabola, and many neuropterids are treated as sensitive indicators for assessing environmental change.

The order Neuroptera is composed of 19 families, presently with 6434 extant species worldwide [4]. Besides the common lacewing groups such as Chrysopidae (green lacewings), Myrmeleontidae (antlions), Hemerobiidae (brown lacewings) and Coniopterygidae (dustywings), the order contains several uncommon or little-known families that have relatively low species diversity and disjunctive distributions. For example, Rhachiberothidae (thorny lacewings), which is similar to Mantispidae with raptorial forelegs, is the smallest family in Neuroptera with only 14 species, all of which are confined to the Afrotropical region. Nevrorthidae (nevrorthid lacewings) have predaceous aquatic larvae, and with 19 species it is the second smallest lacewing family. Further, it is disjunctively distributed in the Mediterranean region, eastern Australia, and East Asia.

The present study focuses on Dilaridae (pleasing lacewings), another little-known lacewing family, with moderate species diversity consisting of 103 extant described species sorted into four genera [5,6,7]. The family has a subglobal distribution due to its absence from the Australian region [5]. Adult dilarids are characterized by sexually dimorphic antennae (flagellum pectinate or thickly filiform in male but slenderly filiform in female), the presence of three ocellus-like tubercles on the head, and the tubular elongate female ovipositor. The biology of Dilaridae is poorly known. Most adult dilarids are typically nocturnal, while some species are diurnal [8]. A distinct disproportion of gender is shown in this family. In general, male dilarids seem to be substantially more abundant and more active than conspecific females since larger numbers of males are collected by light or Malaise traps [8,9,10,11]. However, this phenomenon is at least partly due to the fact that females do not come to the conventionally used traps. The dilarid larvae, which are known to date to represent only seven species in two genera, that is, *Nallachius* Navás, 1909 and *Dilar* Rambur [8,12,13,14,15]. Larvae of both genera are reported to be predatory although they inhabit dead wood or soil [8,12,13,14,15].

The genus *Dilar* Rambur, 1838, is the most species-rich group in Dilaridae, being widespread from the Palearctic to the Oriental region. However, the species diversity of *Dilar* was scarcely explored before the mid-2010s; prior to that only few fragmentary descriptive studies were published [10,11,16,17,18,19,20,21,22]. Owing to extensive taxonomic revisions in the past six years, 33 new species of *Dilar* have been described mainly from Asia, which considerably enhanced our knowledge on the fauna of *Dilar* [6,8,23,24,25,26,27,28,29,30,31]. To date, a total of 75 extant species of *Dilar* have been described [6,7]. The known species diversity of *Dilar* (excluding four doubtful species) can be broadly categorized into the following regions: Europe, North Africa and West Asia, 13 species; Central Asia, 5 species; southern part of South Asia, 6 species; southern Tibet and adjacent area, 3 species; Palearctic part of East Asia, 7 species; Oriental China, 26 species; Indochina Peninsula, 10 species; Indo-Malaysia, 4 species. The Oriental China subregion was considered a diversification center of *Dilar* by Zhang et al. [26] since the region harbors considerably more species than all other regions in Asia. However, due to inadequate field investigations and taxonomic studies [27], the species diversity of *Dilar* may be critically underestimated in regions adjacent to Oriental China, such as Southeast Asia that have habitats similar to those of the Oriental species.

Here we report on 12 new species of *Dilar* from southwestern China and the Indochina Peninsula. The new species display diverse morphological characters in wing marking patterns and male genital sclerites. A first assessment of the worldwide diversity of *Dilar* can now be made based on all available faunal data. Yunnan (southwestern China) emerges as a region with a rich species diversity and notable endemism of the genus.

## 2. Materials and Methods

### 2.1. Taxonomic Study

Specimens were mostly collected by sweeping vegetation during daytime with a net and through the use of light-traps and Malaise traps. The specimens examined in the present study were deposited in the Entomological Museum, China Agricultural University (CAU), Beijing, China; the California Academy of Sciences (CASC), San Francisco, CA, USA; the Natural History Museum Geneva (NHMG), Geneva, Switzerland; the Natural History Museum Vienna, Austria (NMW); the Collection of Hubert Rausch and Renate Rausch (CHRR). Additional information on the examined material is placed in square brackets to expand or augment the frequently cryptic text of the original labels or to provide geographic coordinates.

Genitalic preparations were made by clearing the apex of the abdomen with KOH heated to 120 °C on a hot plate for about 5–6 min. After rinsing the KOH with distilled water, the apex of the abdomen was transferred to glycerin for further examination. Habitus photos of adults were taken by using Nikon D800 and Nikon D850 digital camera with Nikon MICRO NIKKOR 105 mm lens. The photos of genitalia were taken using the Nikon D850 together with a Leica DM 2000 optical microscope, and the genitalic figures were made by line-drawing using a Nikon SMZ745 stereo microscope.

The terminology of wing venation generally follows Liu et al. [5]. Terminology of the genitalia generally follows U. Aspöck and H. Aspöck [32]. The abbreviations for wing veins are: A, anal vein; C, costa; Cu, cubitus; CuA, cubitus anterior; CuP, cubitus posterior; MA, media anterior; MP, media posterior; R, radius; RA, radius anterior; RP, radius posterior; ScP, subcostal posterior.

### 2.2. Species Richness Analysis

Species richness is the number of different species of a particular taxon or life form represented in a particular community or region. It is the simplest, most intuitive and most frequently used measurement of species diversity and features prominently in foundational models of community ecology [33,34]. Here, we summarized collecting information from a total of 818 specimens of 90 extant *Dilar* species (including 83 valid species, four doubtful species which were previously regarded as nomina dubia and three undetermined species) in a Microsoft Excel 2016 datasheet (Appendix A). Data on species richness (Appendix A) were extracted from the original dataset in Appendix A for first-level national administrative divisions (i.e., countries), with the exception of Andorra, Armenia, Bosnia and Herzegovina, Korea, Lebanon, Montenegro, North Macedonia, Sri Lanka and Syria where the count applied to the whole country. In Bursa (Turkey), Hunan (China), and Shanxi (China), undetermined species were distributed together with valid species. In these cases, the species number for each of the areas was determined only by the number of valid species. However, in the following areas, that is, Lebanon, Alger (Algeria), Antalya (Turkey), Kırklareli (Turkey), Dagestan (Russia), Jiangsu (China) and Shandong (China), there are only doubtful or undetermined species records. Thus, the species number for these areas was given as 1 (Appendix A). Localities of distribution records were mainly taken from original descriptions or subsequent literature references for certain species (Appendix A), while the remaining distribution records were obtained from our newly examined specimens of various species deposited in CAU, CHRR, HUAC and NMW (Appendix A). The abbreviations of collections in Appendix A are given in Appendix A. The map of species richness was made by ArcGIS 10.2 based on the species number of each region (Appendix A). The database of country and administrative areas in the map was downloaded from Website GADM version 3.6 (https://gadm.org/data.html, accessed on 1 January 2021). Coordinate information was acquired for the specific location of each specimen via Google Maps @2020. The regional distribution maps were made by ArcGIS 10.2 based on plausible locations of 87 species (including three undetermined species but excluding four doubtful species). Some of location marks are super-imposed due to sympatric distribution. In this case, those marks are respectively processed as a combined mark with corresponding colors by Adobe Photoshop 2018.

To test for any correlation between elevation and latitude in each region, the specimen data with detailed collecting information of altitude and latitude in Appendix A were fitted using linear regression in Microsoft Excel 2016 and SPSS 19.0 (SPSS Inc., Chicago, IL, USA). For altitude data originally counted as a range with drops less than 1000 m, the mean value of the drop data was used for the analysis.

## 3. Results

### 3.1. Taxonomy

Figure 1, Figure 2, Figure 3, Figure 4, Figure 5, Figure 6, Figure 7, Figure 8, Figure 9, Figure 10, Figure 11, Figure 12, Figure 13, Figure 14, Figure 15, Figure 16 and Figure 17. Genus *Dilar* Rambur, 1838.

*Dilar* Rambur, [1838] 1837–1840: pl. 9: Figure 4 and Figure 5. Type species: *Dilar nevadensis* Rambur, [1838] 1837–1840: pl. 9 (monotypy).

*Cladocera* Hagen, 1860: 56. Nomen nudum.

*Lidar* Navás, 1909: 153. Type species: *Dilar meridionalis* Hagen, 1866: 295 (original designation).

*Fuentenus* Navás, 1909: 154. Type species: *Dilar campestris* Navás, 1903: 380 (original designation).

*Rexavius* Navás, [1909] 1908–1909: 664. Type species: *Dilar nietneri* Hagen, 1858: 482 (designated by Navás, 1914: 10).

*Nepal* Navás, [1909>] 1908–1909: 661. Type species: *Nepal harmandi* Navás, [1909] 1908–1909: 661 (original designation).

Description. Male antennae pectinate, but distally 6–8 flagellomeres filiform. Mouthparts short. Wings broad, generally with numerous dark markings. Forewing generally with two nygmata, one at base (between MA + MP) and one middle (between RP + MA); subcostal space with two or more crossveins; MA separating from R distinctly proximad separation of RA and RP; trichosors generally present between most costal crossveins and between marginal forks of RA, RP, MA, MP, CuA and/or CuP. Male genitalia: Tergum 9 in dorsal view with a deeply V- or U-shaped posterior incision, leaving a pair of broad hemitergites, and sometimes with a posteromedial projection (dorsoprocessus). Sternum 9 generally much shorter than tergum 9. Ectoproct highly specialized, generally comprising dorsal and ventral sclerites, posteriorly with projections of various quantity and shape. Complex of gonocoxites 9, 10 and 11 comprising two pairs of sclerites, that is, gonocoxites 9 and 10, and a transverse sclerite, that is, fused gonocoxites 11; fused gonocoxites 11 laterally connecting to bases of gonocoxites 9. Hypandrium internum generally trapezoidal, with lateral margins slightly arcuate. Female genitalia: No sclerotized subgenitale in most species. Tergum 9 generally narrow and strongly extending ventrad in lateral view. Bursa copulatrix with a tubular colleterial gland, a specialized basal part of bursa copulatrix which is usually variously shaped among species, and a pair of bursal accessory glands. Ectoproct small, ovoid.

Distribution. The genus *Dilar* occurs in the Palearctic and Oriental regions, encompassing Europe (Spain, Andorra, Portugal, France including Corsica, Italy including Sardinia, Croatia, Bosnia-Herzegovina, Montenegro, Macedonia, Bulgaria, Kosovo, Albania, Greece including several western and eastern islands, Ukraine, Russia), northern Africa (Algeria, Tunisia), and Asia (Armenia, Turkey, Lebanon, Syria, Iran, Afghanistan, Turkmenistan, Kyrgyzstan, Uzbekistan, Tajikistan, Bhutan, Nepal, Pakistan, India, Sri Lanka, China, Korea, Japan, Myanmar, Laos, Thailand, Vietnam, Malaysia, Indonesia).

#### 3.1.1. *Dilar cangyuanensis* sp. nov.

Figure 1A,B, Figure 3A, Figure 4A–L and 25.

**Figure 1 insects-12-00451-f001:**
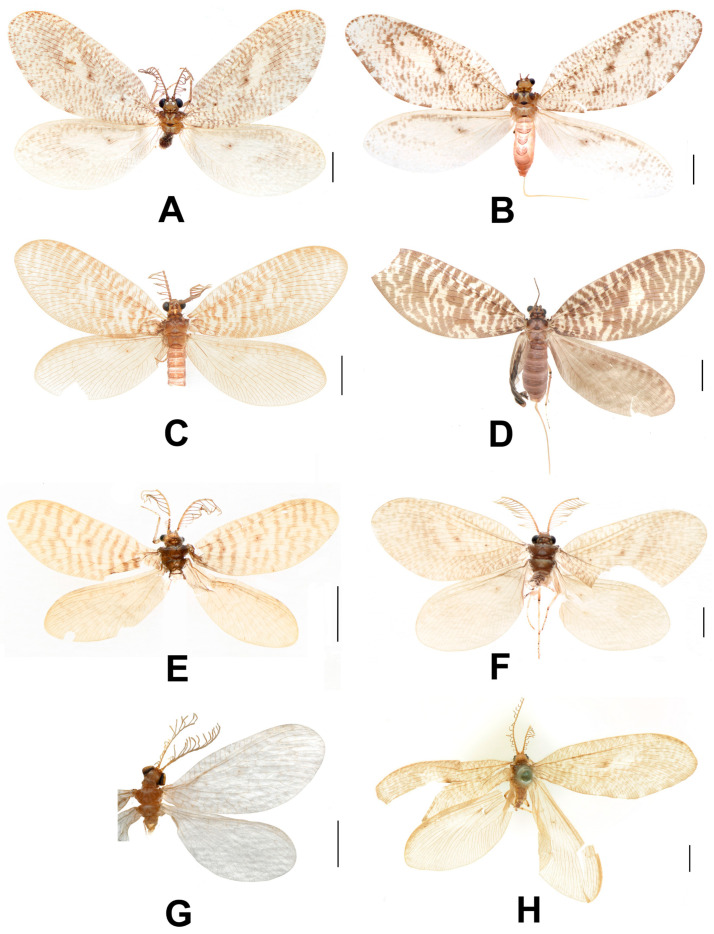
Adults of *Dilar* spp. (**A**) *Dilar cangyuanensis* sp. nov., male holotype; (**B**) *Dilar cangyuanensis* sp. nov., female; (**C**) *Dilar daweishanensis* sp. nov., male holotype; (**D**) *Dilar daweishanensis* sp. nov., female; (**E**) *Dilar forcipatus* sp. nov., male holotype; (**F**) *Dilar laoticus* sp. nov., male holotype; (**G**) *Dilar malickyi* sp. nov., male holotype; (**H**) *Dilar nujianganus* sp. nov., male holotype. Scale bars: 2.0 mm.

**Figure 2 insects-12-00451-f002:**
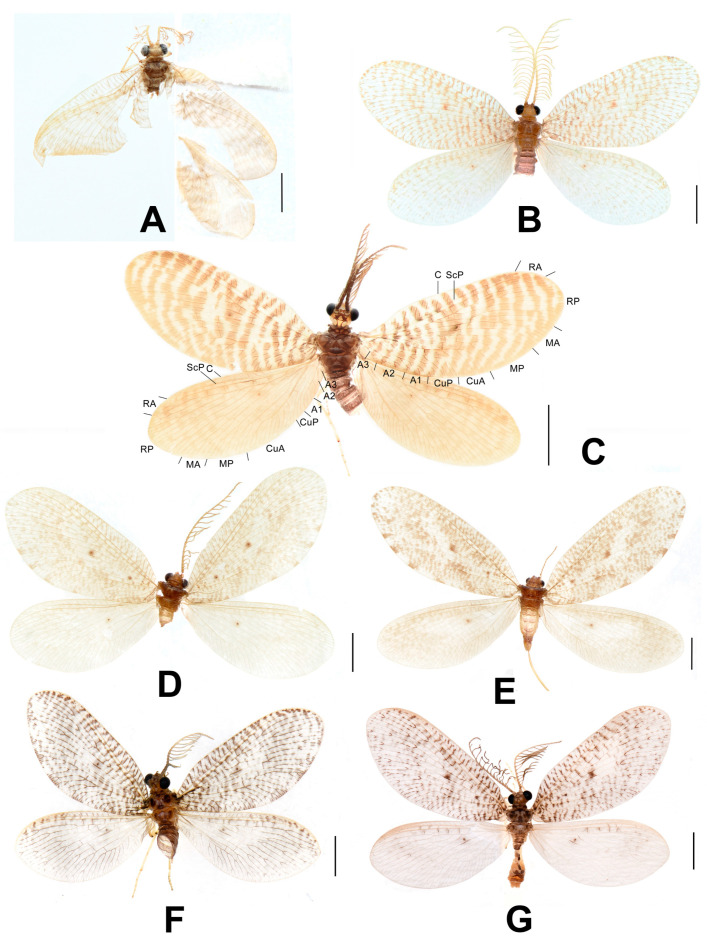
Adults of *Dilar* spp. (**A**) *Dilar phraenus* sp. nov., male holotype; (**B**) *Dilar rauschorum* sp. nov., male holotype; (**C**) *Dilar striatus* sp. nov., male holotype; (**D**) *Dilar weibaoshanensis* sp. nov., male holotype; (**E**) *Dilar weibaoshanensis* sp. nov., female; (**F**) *Dilar yucheni* sp. nov., male holotype; (**G**) *Dilar zhangweiae* sp. nov., male holotype. Scale bars: 2.0 mm.

**Figure 3 insects-12-00451-f003:**
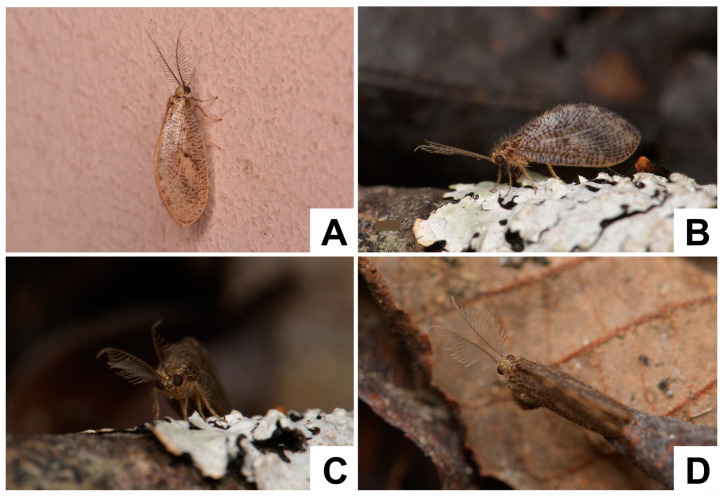
Live adults of *Dilar* spp. (**A**) *Dilar cangyuanensis* sp. nov., male (photograph by Yuchen Zheng); (**B**–**D**) *Dilar yucheni* sp. nov., male (photograph by Yuchen Zheng).

**Figure 4 insects-12-00451-f004:**
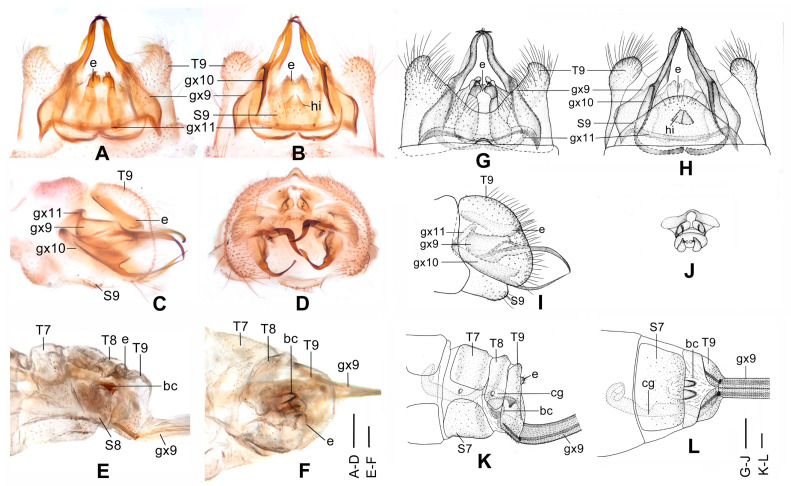
*Dilar cangyuanensis* sp. nov. (**A**–**D**) male holotype, genitalia, photographs: (**A**) dorsal view; (**B**) ventral view; (**C**) lateral view; (**D**) caudal view. (**E**–**F**) female, genitalia, photographs: (**E**) lateral view; (**F**) dorsal view. (**G**–**J**) male holotype, genitalia, line drawings: (**G**) dorsal view; (**H**) ventral view; (**I**) lateral view; (**J**) ectoproct, caudal view. (**K**–**L**) female, genitalia, line drawings: (**K**) lateral view; (**L**) ventral view. bc: bursa copulatrix; cg: colleterial gland; e: ectoproct; gx9: gonocoxite 9; gx10: gonocoxite 10; gx11: fused gonocoxites 11; hi: hypandrium internum; S7–9: sternum 7–9; T7–9: tergum 7–9. Scale bars: 0.2 mm.

Diagnosis. The new species is characterized by the forewing densely speckled and a broad immaculate area present distad from the median nygma, and by the slenderly elongate male gonocoxite 10 with slender base vertically incurved.

Description. Male. Body length 3.8 mm; forewing length 10.2 mm, hindwing length 8.6 mm.

Head generally brown, with light yellow setose tubercles; vertex dark brown. Compound eyes blackish brown. Antenna brown, pedicel distally with a dark brown annular stripe, flagellum with medial branches much longer than those branches at both ends, longest branch nearly 3.0 times as long as corresponding flagellomere, distal 1/3 damaged.

Prothorax brown, pronotum anteriorly and medially with a pair of pale yellow ovoid tubercles respectively; meso- and metathorax yellow, mesonotum brown, darker on anterior, lateral and posterior margins, medially with a pair of light yellow semilunar markings, metanotum paler than mesonotum. Legs generally yellow, but femora, tibiae and each tarsomere dark brown at tip. Wings hyaline, slightly light yellow. Forewing 2.2 times as long as wide, densely speckled, with spots darker at base and middle, a broad immaculate area present distad from the median nygma; three nygmata on left forewing (one at base, two middle), two nygmata on right forewing (base and middle); nygmata surrounded by a brownish spot; longitudinal veins pale brown, interrupted by many brown spots; crossveins pale brown. Hindwing 2.3 times as long as wide, much paler than forewing.

Abdomen brown, each pregenital segment dorsally dark brown. Tergum 9 in dorsal view with an arcuate anterior incision, a nearly U-shaped posterior incision, leaving a pair of subtriangular hemitergites, which are obtuse distally and densely haired. Sternum 9 strongly convex posteriorly, almost half the length of tergum 9 (Figure 4B,H). Ectoproct in dorsal view rectangular, with anterior margin slightly concaved (Figure 4A,G); posterodorsally with a pair of inflated projections and a pair of posteroventrally directed unguiform projections, posteroventrally with a pair of digitiform projections and a pair of unguiform projections (Figure 4D,J). Gonocoxite 9 inflated on proximal half, slenderly elongate, with spinous tip vertically incurved (Figure 4A,G). Gonocoxite 10 slenderly elongate, slightly longer than gonocoxite 9, with a vertically incurved base and spinous tip, submedially with a tiny processus (Figure 4B,H). Fused gonocoxites 11 nearly beam-shaped, with lateral ends angulately curved towards anteriad, laterally connecting to bases of gonocoxites 9 (Figure 4B,H). Hypandrium internum nearly trapezoidal, with lateral margins slightly arcuate (Figure 4B,H).

Female. Body length 6.7 mm; forewing length 12.2 mm, hindwing length 10.3 mm.

Wings with markings much darker than those in male.

Sternum 7 in lateral view nearly rectangular, in ventral view nearly trapezoidal, with truncate posterior margin (Figure 4L). Abdominal segment 8 ventrally without subgenitale. Tergum 9 in lateral view almost as wide as tergum 8 (Figure 4E,K). Bursa copulatrix with colleterial gland tubular and elongate, curved proximally (Figure 4K); basal part of bursa copulatrix in lateral view rounded, posterodorsally with a trumpet-like sclerite (Figure 4E,K), in ventral view present as an ovoid sac, which medially has a pair of U-shaped sclerites (Figure 4L); bursal accessory gland not observed. Ectoproct small, ovoid (Figure 4E,K).

Materials examined. Holotype ♂, China, Yunnan Province, Cangyuan Town, Nangunhe National Nature Reserve, Wengding Station, [23°08′51″ N 99°14′42″ E], 1800 m, 03/V/2019, light, Yuchen Zheng and Hongyu Li (CAU, stored in ETOH). Paratypes 4♂4♀, same data as holotype (CAU, stored in ETOH).

Etymology. The specific epithet “*cangyuanensis*” refers to the type locality of the species. It is an adjective, masculine, nominative, singular as an attribute to the genus name.

Distribution. China (Yunnan).

Remarks. The new species resembles *Dilar lijiangensis* Zhang, Liu, H. Aspöck and U. Aspöck, 2015, in having similar male genital characters, for example, the rectangular ectoproct, the slenderly elongate gonocoxite 9 with inflated base and spinous tip curved medially, the fused gonocoxites 11 with lateral ends angulately curved towards anteriad [26]. However, these two species can be clearly distinguished by the forewing markings (densely spotted in *D. cangyuanensis* sp. nov., but sparsely spotted in *D. lijiangensis*), the shape and length of male gonocoxite 10 (not bifurcated and slightly longer than gonocoxite 9 in *D. cangyuanensis* sp. nov., but submedially bifurcated and much shorter than gonocoxite 9 in *D. lijiangensis*), and the shape of male fused gonocoxites 11 (beam-shaped in *D. cangyuanensis* sp. nov., but nearly W-shaped in *D. lijiangensis*).

#### 3.1.2. *Dilar daweishanensis* sp. nov.

Figure 1C,D, Figure 5A–L and 25.

**Figure 5 insects-12-00451-f005:**
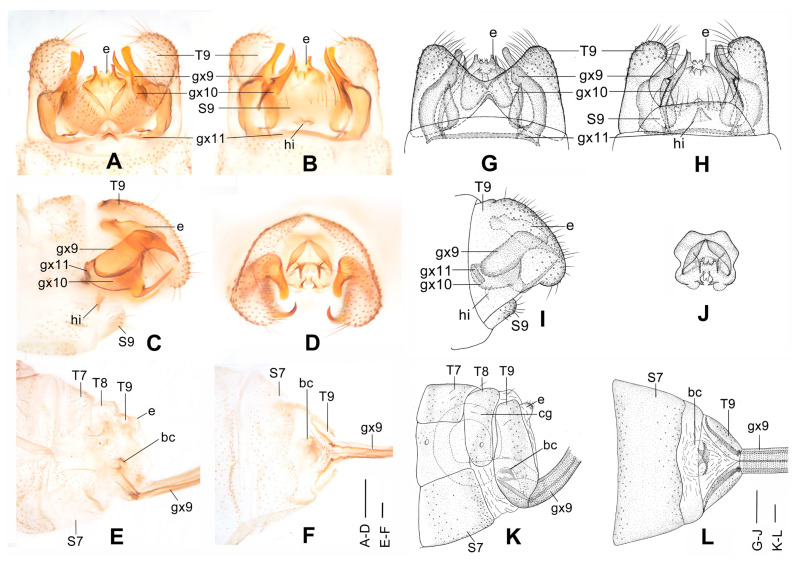
*Dilar daweishanensis* sp. nov. (**A**–**D**) male holotype, genitalia, photographs: (**A**) dorsal view; (**B**) ventral view; (**C**) lateral view; (**D**) caudal view. (**E**–**F**) female, genitalia, photographs: (**E**) lateral view; (**F**) ventral view. (**G**–**J**) male holotype, genitalia, line drawings: (**G**) dorsal view; (**H**) ventral view; (**I**) lateral view; (**J**) ectoproct, caudal view. (**K**–**L**) female, genitalia, line drawings: (**K**) lateral view; (**L**) ventral view. bc: bursa copulatrix; cg: colleterial gland; e: ectoproct; gx9: gonocoxite 9; gx10: gonocoxite 10; gx11: fused gonocoxites 11; hi: hypandrium internum; S7–9: sternum 7–9; T7–9: tergum 7–9. Scale bars: 0.2 mm.

Diagnosis. The new species is characterized by the forewing with dense bands and continuous marking along distal margin, and by the male gonocoxite 9 proximally with a foliate lobe.

Description. Male. Body length 6.2 mm; forewing length 8.8 mm, hindwing length 7.5 mm.

Head generally yellow, with pale brown setose tubercles, tubercles surrounded by dark brown markings; vertex dark brown. Compound eyes blackish brown. Antenna brown, flagellum with medial branches longer than those branches at base, distal half damaged.

Thorax brown; pronotum with anterior and lateral margins slightly darker, mesonotum darker on mesoscutellum as well as along anterior and lateral margins, metanotum paler than mesonotum. Legs generally yellow, but tibiae entirely brown, femora, tibiae and each tarsomere dark brown at tip. Wings transparent, slightly smoky brown. Forewing 2.3 times as long as wide, with dense bands, proximal bands much darker than remaining distal bands, bands connected to each other along distal margin, present as a continuous marking from R to CuP1, a small immaculate area present distad from the median nygma; longitudinal veins pale brown, interrupted by many brown spots; crossveins pale brown. Hindwing 2.3 times as long as wide, immaculate.

Abdomen brown, each pregenital segment dorsally dark brown. Tergum 9 in dorsal view with an arcuate anterior incision, a nearly U-shaped posterior incision, leaving a pair of subtriangular hemitergites, which are obtuse distally and densely haired. Sternum 9 subtrapezoidal, only one third in length of tergum 9, almost truncate posteriad (Figure 5B,H). Ectoproct in dorsal view nearly trapezoidal, with an arcuate anterior incision (Figure 5A,G); posterodorsally with a pair of small semicircular projections and a pair of posteroventrally directed unguiform projections, posteroventrally with a slightly bifid digitiform projection and a pair of bifid unguiform projections (Figure 5D,J). Gonocoxite 9 slenderly elongate, proximally with a foliate lobe connecting to its median portion, distally with laminar tip (Figure 5A,G). Gonocoxite 10 slenderly elongate, almost as long as gonocoxite 9, angulately incurved anteriorly, submedially with a tiny and sharp processus in ventral view, distally with spinous tip (Figure 5B,H). Fused gonocoxites 11 nearly beam-shaped, almost straight, laterally connecting to bases of gonocoxites 9 (Figure 5B,G). Hypandrium internum nearly trapezoidal, with lateral margins slightly arcuate (Figure 5B,H).

Female. Body length 7.2 mm; forewing length 10.8 mm, hindwing length 9.8 mm.

Wings with markings much darker than those in male.

Sternum 7 in lateral view nearly trapezoidal, in ventral view nearly rectangular, with posterior margin slightly concaved at middle (Figure 5E,F,K,L). Segment 8 ventrally without subgenitale. Tergum 9 in lateral view slightly wider than tergum 8 (Figure 5E,K). Bursa copulatrix with colleterial gland tubular and elongate, curled into a circle; basal part of bursa copulatrix present as a round sac in lateral view (Figure 5K) and anteriorly concaved in ventral view, with anterior half membranous and strongly sclerotized posteromedially (Figure 5F,L); bursal accessory gland not observed. Ectoproct small, ovoid (Figure 5E,K).

Materials examined. Holotype ♂, China, Yunnan Province, Pingbian, Daweishan Forest Park, 22°54′24″ N 103°25′48″ E, 1962 m, 16/VII/2016, Yanan Lv (CAU, stored in ETOH). Paratype 1♀, same data as holotype (CAU, stored in ETOH).

Etymology. The specific epithet “*daweishanensis*” refers to the type locality of the new species, that is, Mt. Daweishan in Yunnan Province, China. It is an adjective, masculine, nominative, singular as an attribute to the genus name.

Distribution. China (Yunnan).

Remarks. The new species appears to be related to the species of the *Dilar guangxiensis* species-group through the shared character of the male gonocoxite 10 submedially with a processus connecting to gonocoxite 9. Among the species of the *Dilar guangxiensis* species-group, the new species mostly resembles *D. aspoeckorum* Martins, Flint and Liu, 2018, from northeastern Vietnam in having similar transverse stripes on the forewing, same modifications on the male ectoproct, thin and elongate male gonocoxite 10 submedially with a tiny processus in ventral view, and beam-shaped male fused gonocoxites 11. However, it can be distinguished from the latter species by the absence of dorsoprocessus and the male gonocoxite 9 proximally with a foliate lobe. In *D. aspoeckorum*, the male tergum 9 has a subtriangular dorsoprocessus, and the male gonocoxite 9 lacks an additional lobe proximally [29].

#### 3.1.3. *Dilar forcipatus* sp. nov.

Figure 1E, Figure 6A–H and 26.

**Figure 6 insects-12-00451-f006:**
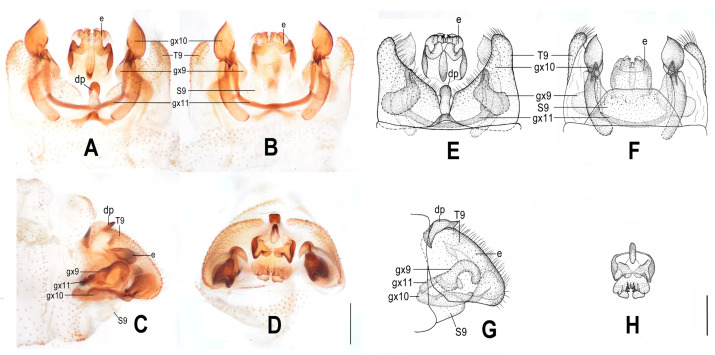
*Dilar forcipatus* sp. nov. (**A**–**D**) male holotype, genitalia, photographs: (**A**) dorsal view; (**B**) ventral view; (**C**) lateral view; (**D**) caudal view. (**E**–**H**) male holotype, genitalia, line drawings: (**E**) dorsal view; (**F**) ventral view; (**G**) lateral view; (**H**) ectoproct, caudal view. dp: dorsoprocessus; e: ectoproct; gx9: gonocoxite 9; gx10: gonocoxite 10; gx11: fused gonocoxites 11; S9: sternum 9; T9: tergum 9. Scale bars: 0.2 m.

Diagnosis. The new species is characterized by the forewing with ca. 12 dark brown transverse bands, by the male ectoproct dorsally with an oblong projection at middle, by the rod-like male gonocoxite 10 with distal one third swollen, and by the male fused gonocoxites 11 with a pair of slenderly elongate lobes near both ends.

Description. Male. Body length 3.5 mm; forewing length 5.6 mm, hindwing length 4.9 mm.

Head generally yellowish brown, with brown setose tubercles; vertex yellow. Compound eyes blackish brown. Antenna generally pale brown, pedicel slightly darker, flagellum with branches slightly darker than flagellomeres, medial branches noticeably/much longer than those branches at both ends, longest branch nearly 2.7 times as long as corresponding flagellomere, distal half damaged.

Thorax brown; pronotum pale brown, medially with a pair of yellow ovoid tubercles; meso- and metanotum darker on anterior and lateral margins. Legs pale brown, tibiae and each tarsomere dark brown at tip. Wings light yellow. Forewing 2.4 times as long as wide, in general with 12 dark brown transverse bands, basal four bands much darker than remaining distal bands, bands on distal half mostly interrupted medially and some of them forked marginally; longitudinal veins pale brown, interrupted by many brown spots; crossveins brown. Hindwing 2.5 times as long as wide, light yellow, much paler than forewing.

Abdomen pale brown. Tergum 9 in dorsal view with an arcuate incision, a nearly U-shaped posterior incision, and a dorsoprocessus, leaving a pair of subtriangular hemitergites, which are obtuse distally and densely setose; dorsoprocessus suboblong, slightly narrowed at middle, strongly sclerotized on distal half (Figure 6A,E). Sternum 9 subtrapezoidal, only half the length of tergum 9 (Figure 6F). Ectoproct in dorsal view rectangular, with an oblong projection at middle, laterally strongly sclerotized (Figure 6A,E); posterodorsally with a pair of inflated projections, posteroventrally with a pair of bifid unguiform projections, a pair of inflated projections and a pair of digitiform projections (Figure 6D,H). Gonocoxite 9 short and inflated, narrowed distally (Figure 6A,E). Gonocoxite 10 nearly twice as long as gonocoxite 9, with pointed tip, proximal two thirds of rod-like but distal one third swollen, joining portion of these two parts strongly sclerotized into a zigzag and a transverse ridge (Figure 6B,F). Fused gonocoxites 11 beam-shaped, slightly convex anteriorly, with a pair of slenderly elongate lobes near both ends, connecting to distal one third of gonocoxites 10, laterally connecting to bases of gonocoxites 9 (Figure 6A,B,E,F). Hypandrium internum lost.

Female. Unknown.

Materials examined. Holotype ♂, Laos, Hua Phan Prov[ince], Ban Saleui, Phou Pan-Mt., 20°13′30″ N 103°59′26″ E, 1350–1900 m, 21/IV/2010, leg. C. Holzschuh + locals (CHRR, stored in ETOH).

Etymology. The specific epithet “*forcipatus*” (=equipped with forceps) refers to the shape of the male gonocoxites 10, which can be likened to a pair of large pincers. It is an adjective, masculine, nominative, singular as an attribute to the genus name.

Distribution. Laos (Hua Phan).

Remarks. The new species appears to be related to the species of the *D. guangxiensis* species-group by the presence of dorsoprocessus on male tergum 9. However, the new species greatly differs from the other known *Dilar* species in several male genital characters, for example, the ectoproct dorsally with an oblong projection at middle, the gonocoxite 10 with rodlike anterior part and a strongly swollen posterior part, joining portion of these two parts strongly sclerotized into a zigzag and a transverse ridge. In addition, the male fused gonocoxites 11 of *D. forcipatus* sp. nov. has a pair of slenderly elongate lobes near both ends, which is only shared by *Dilar harmandi* (Navás, 1909) [24] among all other known species of *Dilar*.

Based on markings on forewing, the new species resembles *Dilar rotundatus* Zhang, Liu and Winterton from Thailand, *Dilar striatus* sp. nov. from Vietnam, and *Dilar yangi* Zhang, Liu, H. Aspöck and U. Aspöck, 2015 from China, in having a similar forewing marking pattern, distinguished by a series of transverse bands through the whole wing.

#### 3.1.4. *Dilar laoticus* sp. nov.

Figure 1F, Figure 7A–H and 26.

**Figure 7 insects-12-00451-f007:**
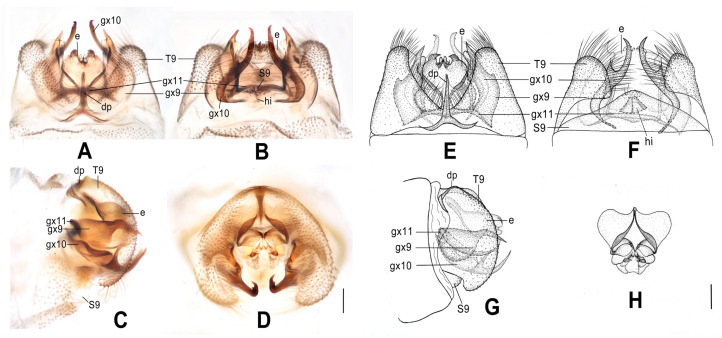
*Dilar laoticus* sp. nov. (**A**–**D**) male holotype, genitalia, photographs: (**A**) dorsal view; (**B**) ventral view; (**C**) lateral view; (**D**) caudal view. (**E**–**H**) male holotype, genitalia, line drawings: (**E**) dorsal view; (**F**) ventral view; (**G**) lateral view; (**H**) ectoproct, caudal view. dp: dorsoprocessus; e: ectoproct; gx9: gonocoxite 9; gx10: gonocoxite 10; gx11: fused gonocoxites 11; hi: hypandrium internum; S9: sternum 9; T9: tergum 9. Scale bars: 0.2 mm.

Diagnosis. The new species is characterized by the forewing with many brown spots mostly connected with each other and a broad immaculate area distad from the median nygma, the male gonocoxite 9 subdistally swollen with a narrowly elongate lobe connecting to fused gonocoxites 11, and the male gonocoxite 10 subdistally with a tiny pointed processus.

Description. Male. Body length 5.1 mm; forewing length 10.6 mm, hindwing length 9.1 mm.

Head generally brown, with yellowish brown setose tubercles; vertex brown. Compound eyes blackish brown. Antenna generally pale brown, pedicel distally with a dark brown annular stripe, flagellum with medial branches longer than those branches at base, longest branch nearly 3.0 times as long as corresponding flagellomere, distal half damaged.

Thorax brown; pronotum medially with a pair of pale brown ovoid tubercles, meso- and metanotum pale brown and much paler medially. Legs generally yellow, with femora, tibiae and each tarsomere dark brown at tip. Wings pale smoky brown. Forewing 2.1 times as long as wide, with many brown spots, which are mostly connected with each other and arranged as discontinuous stripes; spots darker at base and middle, a broad immaculate area present distad from the median nygma, median nygma surrounded by a brown spot; longitudinal veins light yellow, interrupted by numerous brown spots; crossveins light yellow. Hindwing 2.0 times as long as wide, much paler than forewing.

Abdomen yellowish brown. Tergum 9 in dorsal view with an arcuate incision, a nearly V-shaped posterior incision and a dorsoprocessus, leaving a pair of subtriangular hemitergites, which are obtuse distally and densely setose; dorsoprocessus narrowly elongate and tapering, with median region strongly sclerotized forming a longitudinal ridge (Figure 7A,E). Sternum 9 convex posteriorly, only half the length of tergum 9 (Figure 7B,F). Ectoproct in dorsal view nearly trapezoidal, with an arcuate anterior incision (Figure 7A,E); posterodorsally with a pair of posteromedially directed unguiform projections; posteroventrally with a pair of unguiform projections and a pair of digitiform projections (Figure 7D,H). Gonocoxite 9 inflated on proximal half, medially with a tiny projection on lateral margin, distal half tapering, with unguiform tip, subdistally swollen on inner margin, extended into a narrowly elongate lobe connecting to fused gonocoxites 11 (Figure 7A,E). Gonocoxite 10 slenderly elongate, slightly longer than gonocoxite 9, with angulately incurved base and unguiform tip, submedially with a falcate projection on outer margin, subdistally with a tiny and pointed processus (Figure 7B,F). Fused gonocoxites 11 beam-shaped, almost straight, but slightly prominent anteromedially, laterally curved anteriad, connecting to bases of gonocoxites 9 (Figure 7B,F). Hypandrium internum nearly trapezoidal, with lateral margins slightly arcuate (Figure 7B,F).

Female. Unknown.

Materials examined. Holotype ♂, Laos, Prov[ince] Hua Phan, Phou Pan (M[oun]t[ain].), Ban Saleui (Ort), 20°13′30″ N 103°59′26″ E, 1350–1900 m MSL, 01–15/05[V]/2010, leg. C. Holzschuh (CHRR, stored in ETOH). Paratype 1♂, same data as holotype (NMW, stored in ETOH).

Etymology. The specific epithet “*laoticus*” refers to the country, where the species has been discovered. It is an adjective, masculine, nominative, singular as an attribute to the genus name. 

Distribution. Laos (Hua Phan).

Remarks. The new species represents the first record of the genus *Dilar* from Laos. It is a member of *D. guangxiensis* species-group based on the presence of dorsoprocessus on male tergum 9 and the male gonocoxite 10 submedially with a pointed projection connecting to gonocoxite 9. With respect to the species of the *D. guangxiensis* species-group, the new species appears to be closely related to *D. aspoeckorum* based on the presence of a broad immaculate area on forewing, the proximally inflated male gonocoxite 9 with pointed tip and the slenderly elongate male gonocoxite 10 with angulately incurved base and spinous tip. However, it can be distinguished from the latter species by the male ectoproct posteroventrally with only a pair of unguiform projections (two pairs of unguiform projections present in *D. aspoeckorum*), the male gonocoxite 9 subdistally swollen and with a lobe connecting to fused gonocoxites 11 (subdistally without additional lobe in *D. aspoeckorum*), the male gonocoxite 10 subdistally with a tiny processus on outer margin (subdistally without additional processus in *D. aspoeckorum*), and the male fused gonocoxites 11 slightly prominent anteromedially (fused gonocoxite 11 not prominent at middle in *D. aspoeckorum*). Moreover, the markings along distal section of costal space are also clearly different: distinctly dark in *D. aspoeckorum* and invisible in *D. laoticus* sp. nov. [29].

#### 3.1.5. *Dilar malickyi* sp. nov.

Figure 1G, Figure 8A–H and 26.

**Figure 8 insects-12-00451-f008:**
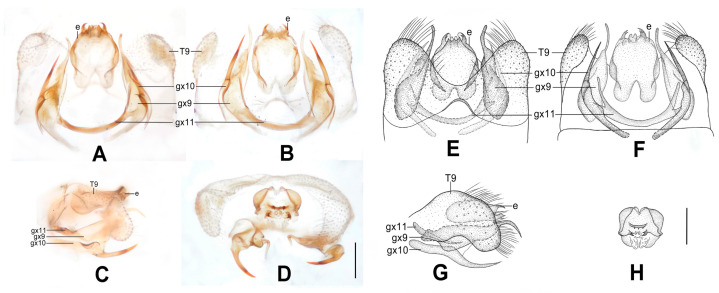
*Dilar malickyi* sp. nov. (**A**–**D**) male holotype, genitalia, photographs: (**A**) dorsal view; (**B**) ventral view; (**C**) lateral view; (**D**) caudal view. (**E**–**H**) male holotype, genitalia, line drawings: (**E**) dorsal view; (**F**) ventral view; (**G**) lateral view; (**H**) ectoproct, caudal view. e: ectoproct; gx9: gonocoxite 9; gx10: gonocoxite 10; gx11: fused gonocoxites 11; T9: tergum 9. Scale bars: 0.2 mm.

Diagnosis. The new species is characterized by the sparsely spotted pale brown forewing, the male ectoproct posterodorsally with three pointed processes of same length, the male gonocoxite 9 with narrowly elongate tip, and the fused male gonocoxites 11 obviously bifurcated at both ends.

Description. Male. Body length 4.5 mm; forewing length 6.2 mm, hindwing length 4.4 mm.

Head yellow, with yellow setose tubercles; vertex yellow. Compound eyes blackish brown. Antenna generally light yellow, flagellum with each branch on medial flagellomere darkened at middle but much paler on proximal and distal portions, medial branches longer than those branches at base, longest branch nearly 4.0 times as long as corresponding flagellomere, distal 1/3 damaged.

Thorax yellow; pronotum light yellow, meso- and metanotum slightly darker. Legs yellow, with tibiae slightly darker at tip. Wings hyaline. Forewing 2.2 times as long as wide, sparsely spotted, markings very pale and indistinct; two nygmata present, one at base and one middle; longitudinal veins light yellow, interrupted by numerous pale brown spots; crossveins light yellow. Hindwing 2.0 times as long as wide, immaculate.

Abdomen pale brown. Tergum 9 in dorsal view with an arcuate anterior incision, and a nearly U-shaped posterior incision, leaving a pair of subtriangular hemitergites, which are obtuse distally and densely haired. Sternum 9 completely damaged and lost. Ectoproct in dorsal view nearly rectangular, with an arcuate anterior incision (Figure 8A,E); posterodorsally with three pointed processes, which are nearly equal in length, posteroventrally with a pair of short and inflated projections, a pair of bifid unguiform projections and a pair of digitiform projections (Figure 8D,H). Gonocoxite 9 slenderly elongate, with inflated base and narrowly elongate tip (Figure 8A,E). Gonocoxite 10 almost as long as gonocoxite 9, with incurved base and sclerotized spinous tip (Figure 8B,F). Fused gonocoxites 11 nearly beam-shaped, anteriorly convex, bifurcated at both ends, with bifurcation connecting to proximal and median portions of gonocoxites 9 (Figure 8A,B,E,F). Hypandrium internum lost.

Female. Unknown.

Materials examined. Holotype ♂, Thailand, [Mae Hong Son, Pang Mapha District] Tham Than Lod N[ational]P[ark], 14.46° N/99.20° E [14°27′ N, 99°12′ E], 500 m, 5/IV/1989, Malicky and Wanleclaq/K7 (CHRR, stored in ETOH).

Etymology. The new species is dedicated to Prof. Dr. Hans Malicky, who collected the holotype of the new species. It is a substantive, masculine, genitive, singular of the latinized form of Malicky as an attribute to the genus name.

Distribution. Thailand (Mae Hong Son).

Remarks. The new species is distantly related to *D. loeinensis* Zhang, Liu and Winterton, 2016, from Thailand in having similar male genital characters, for example, modifications on ectoproct, proximally inflated gonocoxite 9, strongly incurved gonocoxite 10 and fused gonocoxites 11 expanded at both ends. However, *D. malickyi* sp. nov. has slender distal portion of male gonocoxite 9, slenderly elongate male gonocoxite 10 without median projection, and sublateral portion of male fused gonocoxites 11 strongly bifurcated. By contrast, in *D. loeinensis* the distal portion of male gonocoxite 9 is tapering, the median portion of male gonocoxite 10 bears a pointed projection, and the sublateral portion of male fused gonocoxites 11 is slightly inflated but not bifurcated [27].

#### 3.1.6. *Dilar nujianganus* sp. nov.

Figure 1H, Figure 9A–H and 25.

**Figure 9 insects-12-00451-f009:**
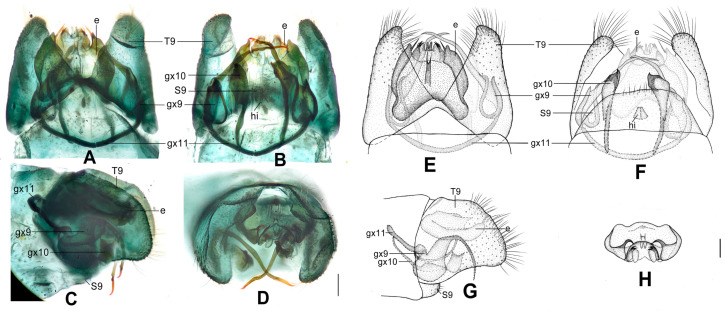
*Dilar nujianganus* sp. nov. (**A**–**D**) male holotype, genitalia, photographs: (**A**) dorsal view; (**B**) ventral view; (**C**) lateral view; (**D**) caudal view. (**E**–**H**) male holotype, genitalia, line drawings: (**E**) dorsal view; (**F**) ventral view; (**G**) lateral view; (**H**) ectoproct, caudal view. e: ectoproct; gx9: gonocoxite 9; gx10: gonocoxite 10; gx11: fused gonocoxites 11; hi: hypandrium internum; S9: sternum 9; T9: tergum 9. Scale bars: 0.2 mm.

Diagnosis. The new species is characterized by the densely spotted forewing with a continuously darkened distal section of costal margin and with an immaculate area present distad from the median nygma, the large male ectoproct with a projection deeply bifurcated at middle on dorsal sclerite, the slenderly elongate male gonocoxite 9 proximal 1/3 modified into a loop, and the male gonocoxite 10 with proximal half rodlike and distal half inflated.

Description. Male. Body length 5.8 mm; forewing length 12.8 mm, hindwing length 10.8 mm.

Head generally brown, with yellowish brown setose tubercles; vertex brown. Compound eyes blackish brown. Antenna pale brown, flagellum with medial branches longer than those branches at base, longest branch nearly 3.0 times as long as corresponding flagellomere, distal half damaged.

Prothorax brown, pronotum medially with a pair of pale brown ovoid tubercles; meso- and metathorax yellow, mesonotum brown, darker on anterior margin, metanotum paler than mesonotum, darker on lateral margin. Legs yellow, but tibiae brown, femora dark brown at tip. Wings light yellow. Forewing 2.6 times as long as wide, densely spotted, spots mostly expanded and fused with each other, continuous along distal section of costal space and distal margin from RA to RP; markings darker at base and along distal section of costal space, an immaculate area present distad from the median nygma; median nygma surrounded by a brown spot; longitudinal veins light yellow, interrupted by numerous brown spots; crossveins light yellow. Hindwing 2.3 times as long as wide, immaculate.

Abdomen pale brown. Tergum 9 in dorsal view with an arcuate anterior incision, a nearly V-shaped posterior incision, leaving a pair of subtriangular hemitergites, which are obtuse distally and densely haired. Sternum 9 subtrapezoidal, slightly less than half the length of tergum 9 (Figure 9B,F). Ectoproct large, in dorsal view with an arcuate anterior incision, dorsal sclerite with a deeply bifurcated projection at middle, extended into a pair of spinous elongate projections (Figure 9A,E); posterodorsally with a pair of posteromedially spinous projections, posteroventrally with a pair of bifid unguiform projections, a pair of inflated projections and a pair of digitiform projections (Figure 9D,H). Gonocoxite 9 slenderly elongate, proximally with narrow lobe, fused at proximal 1/3 into a loop, distally with slender and spinous tip (Figure 9A,B,E,F). Gonocoxite 10 rodlike on proximal half, inflated on distal half, with a pointed tip (Figure 9B,F). Fused gonocoxites 11 nearly U-shaped, laterally connected to bases of gonocoxites 9 (Figure 9A,B,E,F). Hypandrium internum subtrapezoidal, with lateral margins slightly arcuate (Figure 9B,F).

Female. Unknown.

Materials examined. Holotype ♂, China Yunnan Province, Mt. Gaoligongshan, Nujiang Prefecture, Nujiang State Nature Reserve, No. 12 Bridge Camp area, 16.3 air km W of Gongshan, 27.71503° N 98.50244° E [27°42′54″ N 98°30′08″ E], 2775 m, 15–19/VII/2000, malaise trap, Stop#00-23E, D. H. Kavanaugh, C. E. Gnswold, Liang H. B., D. Ubick and Dong D. Z. (CASC, pinned specimen).

Etymology. The specific epithet “*nujianganus*” refers to the type locality of the new species, that is, Nujiang Prefecture, Yunnan Province, China. It is an adjective, masculine, nominative, singular as an attribute to the genus name.

Distribution. China (Yunnan).

Remarks. The new species is distinctly different from the other species of *Dilar* based on the male genital characters, for example, the dorsal sclerite of the large ectoproct medially with a deeply bifurcated projection, and the slenderly elongate gonocoxite 9 proximal 1/3 modified into a loop. In addition, it is notable that the forewing in the new species has a continuously darkened distal section of costal margin, which resembles a pterostigma. In the other known species of *Dilar*, only *D. aspoeckorum* has this character [29].

#### 3.1.7. *Dilar phraenus* sp. nov.

Figure 2A, Figure 10A–H and 26.

**Figure 10 insects-12-00451-f010:**
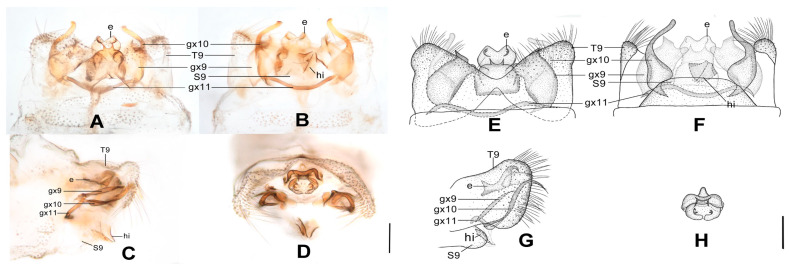
*Dilar phraenus* sp. nov. (**A**–**D**) male holotype, genitalia, photographs: (**A**) dorsal view; (**B**) ventral view; (**C**) lateral view; (**D**) caudal view. (**E**–**H**) male holotype, genitalia, line drawings: (**E**) dorsal view; (**F**) ventral view; (**G**) lateral view; (**H**) ectoproct, caudal view. dp: dorsoprocessus; e: ectoproct; gx9: gonocoxite 9; gx10: gonocoxite 10; gx11: fused gonocoxites 11; S9: sternum 9; T9: tergum 9. Scale bars: 0.2 mm.

Diagnosis. This species is characterized by the forewing with many brownish transversely arcuate stripes and with a broad immaculate area present distad from the median nygma, and by the gonocoxite 9 posteriorly bilobed, comprising a short, broad, subtriangular lobe directed laterally, and a smaller obtuse lobe directed medially.

Description. Male. Body length 3.8 mm; forewing length 7.2 mm, hindwing length 6.5 mm.

Head pale brown, with yellow setose tubercles; vertex brown. Compound eyes blackish brown. Antenna generally yellow, but pedicel brown, flagellum with medial branches much longer than those branches at both ends, longest branch nearly 5.6 times as long as corresponding flagellomere, distal seven flagellomeres simple.

Prothorax pale brown, pronotum yellow, slightly darker at middle, medially with a pair of light yellow ovoid tubercles; meso- and metathorax brown, mesonotum darkened at middle as well as along anterior and lateral margins, metanotum paler than mesonotum. Legs generally yellow, with femora, tibiae and each tarsomere dark brown at tip. Wings hyaline, slightly smoky brown. Forewing almost 2.2 times as long as wide, with many transversely arcuate stripes, a broad immaculate area present distad from the median nygma; two nygmata present, one at base and one middle, median nygma surrounded by a brown spot; longitudinal veins light yellow, interrupted by many brown spots; crossveins pale brown. Hindwing 2.2 times as long as wide, immaculate.

Abdomen pale brown. Tergum 9 in dorsal view with an arcuate anterior incision, and a nearly U-shaped posterior incision, leaving a pair of subtriangular hemitergites, which are obtuse distally and densely haired. Sternum 9 subtrapezoidal, membranous, only half the length of tergum 9 (Figure 10F). Ectoproct in dorsal view nearly rectangular, with anterior margin almost truncate (Figure 10A,E); posterodorsally with a pair of sharply pointed projections directed posteriad, posteroventrally with a pair of bifid unguiform projections (Figure 10D,H). Gonocoxite 9 inflated, posteriorly bilobed, with a short, broad, subtriangular lobe directed laterally and a smaller obtuse lobe directed medially (Figure 10E). Gonocoxite 10 longer than gonocoxite 9, swollen on proximal half, with medially directed subtriangular base; narrowed and strongly sclerotized on distal half, with slenderly elongate laminar tip (Figure 10B,F). Fused gonocoxites 11 nearly W-shaped, laterally connecting to bases of gonocoxites 9 (Figure 10A,B,E,F). Hypandrium internum large, nearly trapezoidal (Figure 10B,F).

Female. Unknown.

Materials examined. Holotype ♂, Thailand, Phrae Province, Wieng Ko Sai National Park, upper Huai Panjane, 17.56° N, 99.34° E [17°33′36″ N 99°20′24″ E], 295 m, 22–23/VI/2002, B[lack]L[ight]T[rap], CMU TEAM (CASC, pinned specimen). Paratypes 2♂, same data as holotype (CASC, pinned specimens).

Etymology. The specific epithet “*phraenus*” refers to the type locality of the new species, that is, Phrae Province, northern Thailand. It is an adjective, masculine, nominative, singular as an attribute to the genus name.

Distribution. Thailand (Phrae).

Remarks. The new species appears to be closely related to *D. truncatus* Li, U. Aspöck, H. Aspöck and Liu, 2020 from Sri Lanka, in having similar male genital characters, for example, the gonocoxite 9 posteriorly truncate and with an obtuse lobe directed medially, the gonocoxites 10 inflated on proximal half and narrowed on distal half, and the W-shaped fused gonocoxites 11. However, it can be distinguished from the latter species by the rectangular male ectoproct and the male gonocoxites 10 with pointed base and laminar tip. In *D. truncatus*, the male ectoproct is nearly subtriangular and the male gonocoxite 10 has obtuse base and spinous tip [31].

#### 3.1.8. *Dilar rauschorum* sp. nov.

Figure 2B, Figure 11A–H and 26.

**Figure 11 insects-12-00451-f011:**
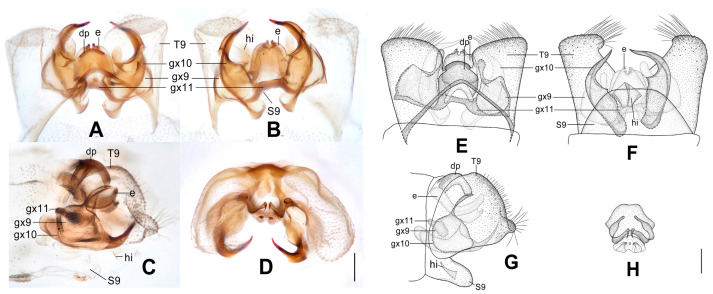
*Dilar rauschorum* sp. nov. (**A**–**D**) male holotype, genitalia, photographs: (**A**) dorsal view; (**B**) ventral view; (**C**) lateral view; (**D**) caudal view. (**E**–**H**) male holotype, genitalia, line drawings: (**E**) dorsal view; (**F**) ventral view; (**G**) lateral view; (**H**) ectoproct, caudal view. dp: dorsoprocessus; e: ectoproct; gx9: gonocoxite 9; gx10: gonocoxite 10; gx11: fused gonocoxites 11; hi: hypandrium internum; S9: sternum 9; T9: tergum 9. Scale bars: 0.2 mm.

Diagnosis. The new species is characterized by the forewing with many short stripes arranged as a transversely arcuate pattern, the pentagonal male dorsoprocessus, and the strongly inflated, rectangular male gonocoxite 9 distally with digitiform tip.

Description. Male. Body length 4.7 mm; forewing length 9.1 mm, hindwing length 8.0 mm.

Head yellowish brown, with light yellow setose tubercles; vertex slightly darker. Compound eyes blackish brown. Antenna generally yellow, but scape brown, pedicel distally with a brown annular stripe, flagellum with medial branches much longer than those branches at both ends, longest branch nearly 5.2 times as long as corresponding flagellomere, distal seven flagellomeres simple.

Prothorax pale brown, pronotum paler on lateral margins, medially with a pair of light yellow ovoid tubercles; meso- and metathorax brown, mesonotum medially with a pair of yellow, semilunar markings, metanotum paler than mesonotum. Legs light yellow, femora, tibiae and each tarsomere dark brown at tip. Wings hyaline, slightly light yellow. Forewing 2.1 times as long as wide, with many short, brown stripes arranged as transversely arcuate patterns; two nygmata present, one at base and one middle, both nygmata surrounded by a brownish spot; longitudinal veins light yellow, interrupted by many brown spots; crossveins brown. Hindwing 2.1 times as long as wide, much paler than forewing.

Abdomen brown. Tergum 9 in dorsal view with an arcuate anterior incision, a nearly U-shaped posterior incision, and a dorsoprocessus, leaving a pair of rectangular hemitergites, which are obtuse distally and densely setose; dorsoprocessus nearly pentagonal, with concaved anterior margin and sclerotized, convex posterior margin (Figure 11A,E). Sternum 9 subtrapezoidal, only half the length of tergum 9 (Figure 11B,F). Ectoproct in dorsal view rectangular, with an arcuate anterior incision (Figure 11A); posterodorsally with a pair of posteroventrally directed unguiform projections, posteroventrally with an oblong projection and a pair of bifid unguiform projections (Figure 11D,H). Gonocoxite 9 short and inflated, nearly rectangular, distally extended into a digitiform projection directed posteriad (Figure 11A,E). Gonocoxite 10 almost twice as long as gonocoxite 9, slenderly elongate, inflated on proximal half and narrowed on distal half, with spinous tip (Figure 11B,F). Fused gonocoxites 11 nearly beam-shaped, laterally curved anteriorly, connecting to bases of gonocoxites 9 (Figure 11B,E). Hypandrium internum large, nearly trapezoidal, with lateral margins slightly arcuate (Figure 11B,F).

Female. Unknown.

Materials examined. Holotype ♂, Thailand, Prov[ince] Mae Hong Son, S[outh]E[ast] Passhöhe zwischen Pai und Soppong [pass summit between Pai Town and Soppong Village], Lichtfang [light], 19°26′17″ N 98°19′53″ E, 1250 m, 22/04[IV]/2000, (30/2000), H. and R. Rausch leg./Thailand 30/2000/K6 (CHRR, stored in ETOH).

Etymology. The species is dedicated to the collectors of the holotype, Hubert and Renate Rausch, who kindly provided specimens of Dilaridae for our studies. The specific epithet “*rauschorum*” is a substantive, genitive, plural of the latinized form of “Rausch” and an attribute to the genus name.

Distribution. Thailand (Mae Hong Son).

Remarks. The new species appears to be related to the species of the *D. guangxiensis* species-group by the presence of dorsoprocessus on male tergum 9. However, it can be distinguished from all species of the *D. guangxiensis* species-group based on the shape of male dorsoprocessus (pentagonal in *D. rauschorum* sp. nov., but subtriangular or subrectangular in species of *D. guangxiensis* species-group), the shape of proximal half of male gonocoxite 9 (strongly inflated, rectangular in *D. rauschorum* sp. nov., but relatively slightly inflated, generally suboblong or subtriangular in species of *D. guangxiensis* species-group), and the male gonocoxites 10 (medially not connecting to gonocoxite 9 in *D. rauschorum* sp. nov., but with a pointy lobe connecting to gonocoxite 9 in species of *Dilar guangxiensis* species-group).

#### 3.1.9. *Dilar striatus* sp. nov.

Figure 2C, Figure 12A–H and 26.

**Figure 12 insects-12-00451-f012:**
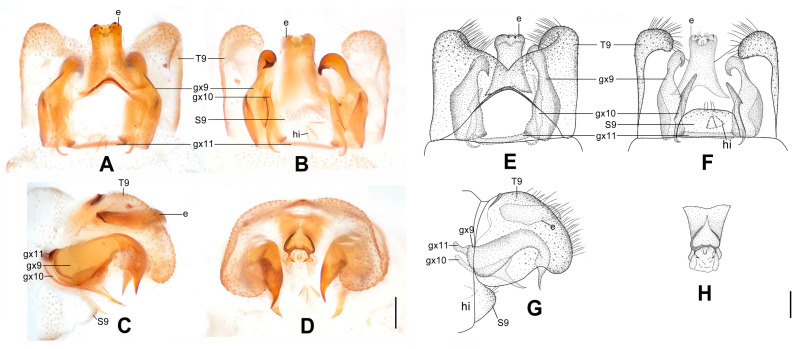
*Dilar striatus* sp. nov. (**A**–**D**) male holotype, genitalia, photographs: (**A**) dorsal view; (**B**) ventral view; (**C**) lateral view; (**D**) caudal view. (**E**–**H**) male holotype, genitalia, line drawings: (**E**) dorsal view; (**F**) ventral view; (**G**) lateral view; (**H**) ectoproct, caudal view. e: ectoproct; gx9: gonocoxite 9; gx10: gonocoxite 10; gx11: fused gonocoxites 11; hi: hypandrium internum; S9: sternum 9; T9: tergum 9. Scale bars: 0.2 mm.

Diagnosis. The new species is characterized by the forewing with nearly 15 dark brown transverse bands and continuously darkened along distal margin from RP3 to CuP, the longitudinally elongate, rectangular male ectoproct, and the inflated male gonocoxite 9 with unguiform tip strongly curved anteriad.

Description. Male. Body length 4.3 mm; forewing length 6.8 mm, hindwing length 6.0 mm.

Head generally yellow, with light yellow setose tubercles, which are surrounded by brown markings; vertex yellow. Compound eyes blackish brown. Antenna generally brown, but scape light yellow, medially with a brown annular stripe, pedicel pale brown, flagellum with branches slightly paler than flagellomeres, medial branches much longer than those branches at both ends, longest branch nearly 4.0 times as long as corresponding flagellomere, distal seven flagellomeres simple.

Thorax brown; pronotum medially with a pair of pale brown ovoid markings; meso- and metanotum paler on lateral margins. Legs light yellow, femora, tibiae and each tarsomere dark brown at tip. Wings light yellow. Forewing 2.4 times as long as wide, with nearly 15 dark brown, transverse bands, which are darker on proximal half and costal space, bands mostly interrupted medially on proximal half, fused together arranged as continuous marking along distal margin from RP3 to CuP; two nygmata present, one at base and one middle; longitudinal veins pale brown, interrupted by many brown spots; crossveins brown. Hindwing 2.4 times as long as wide, much paler than forewing.

Abdomen pale brown. Tergum 9 in dorsal view with an arcuate anterior incision, and a nearly V-shaped posterior incision, leaving a pair of subtriangular hemitergites, which are obtuse distally and densely setose. Sternum 9 nearly trapezoidal, only one third in length of tergum 9 (Figure 12B,F). Ectoproct in dorsal view nearly rectangular, longitudinally elongate, with a slightly arcuate anterior incision (Figure 12A,E); posterodorsally with a pair of semicircular projections and a pair of ventrally directed unguiform projections; posteroventrally with a digitiform projection and a pair of unguiform projections (Figure 12D,H). Gonocoxite 9 inflated on proximal half and tapering on distal half, with strongly curved unguiform tip directed anteriad (Figure 12B,E). Gonocoxite 10 shorter than gonocoxite 9, with incurved, pointed base and tapering, laminar tip, submedially with a tiny and sharp processus connecting to gonocoxite 9 (Figure 12B,F). Fused gonocoxites 11 beam-shaped, almost straight, laterally connecting to bases of gonocoxites 9 (Figure 12A–D). Hypandrium internum nearly trapezoidal, with lateral margins slightly arcuate (Figure 12B,F).

Female. Unknown.

Materials examined. Holotype ♂, Vietnam, Ha Noi Prov[ince], Ba Vi Distr[ict], Mt. Ba Vi, 21°03′35″ N 105°22′02″ E, 1000–1070 m (evergreen forest), 16–18/V/2012, leg. P. Schwendinger and A. Schulz., VN-12/05c (NHMG, stored in ETOH).

Etymology. The specific epithet “*striatus*” (=striated) refers to the marking pattern on forewing, which is accompanied by many brown transverse bands. It is an adjective, masculine, nominative, singular as an attribute to the genus name.

Distribution. Vietnam (Ha Noi).

Remarks. The new species resembles *D. forcipatus* sp. nov., from Laos, *D. rotundatus* from Thailand, and *D. yangi* from China in having similar forewing marking patterns, characterized by a series of transverse bands throughout the whole wing. Based on male genitalia, the new species appears to be somewhat related to the species of the *D. guangxiensis* species-group by the male gonocoxite 10 submedially with a processus connecting to gonocoxite 9. However, the dorsoprocessus on male tergum 9 is present in all species of the *D. guangxiensis* species-group, while it is absent in *D. striatus* sp. nov.

#### 3.1.10. *Dilar weibaoshanensis* sp. nov.

Figure 2D,E, Figure 13A–L and 25.

**Figure 13 insects-12-00451-f013:**
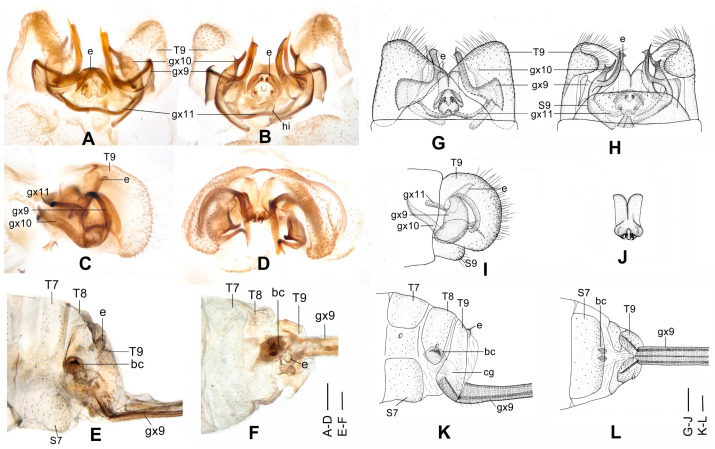
*Dilar weibaoshanensis* sp. nov. (**A**–**D**) male holotype, genitalia, photographs: (**A**) dorsal view; (**B**) ventral view; (**C**) lateral view; (**D**) caudal view. (**E**–**F**) female, genitalia, photographs: (**E**) lateral view; (**F**) dorsal view. (**G**–**J**) male holotype, genitalia, line drawings: (**G**) dorsal view; (**H**) ventral view; (**I**) lateral view; (**J**) ectoproct, caudal view. (**K**–**L**) female, genitalia, line drawings: (**K**) lateral view; (**L**) ventral view. bc: bursa copulatrix; cg: colleterial gland; e: ectoproct; gx9: gonocoxite 9; gx10: gonocoxite 10; gx11: fused gonocoxites 11; hi: hypandrium internum; S7–9: sternum 7–9; T7–9: tergum 7–9. Scale bars: 0.2 mm.

Diagnosis. The new species is characterized by the densely speckled forewing and an immaculate area distad the median nygma, and by the slenderly elongate male gonocoxite 10 bifurcated at tip and submedially with an additional slender, falcate lobe.

Description. Male. Body length 4.5 mm; forewing length 9.8 mm, hindwing length 8.5 mm.

Head generally pale brown, with yellowish brown setose tubercles; vertex brown. Compound eyes blackish brown. Antenna pale brown, pedicel distally with a dark brown stripe, flagellum with medial branches much longer than those on both ends, longest branch nearly 3.0 times as long as corresponding flagellomere, distal six flagellomeres simple.

Prothorax brown, pronotum medially with a pair of pale brown ovoid markings; meso- and metathorax brown, mesonotum darker on mesoscutellum as well as along anterior and lateral margins, metanotum paler than mesonotum. Legs pale brown with brown setae, femora dark brown at tip. Wings light yellow. Forewing 2.3 times as long as wide, densely speckled, spots mostly expanded and fused with each other; markings on proximal and median areas slightly darker, an immaculate area present distad from median nygma; two nygmata present, one at base and one middle, both nygmata surrounded by a large brownish spot; longitudinal veins pale brown, interrupted by many brown spots; crossveins pale brown. Hindwing 2.0 times as long as wide, almost immaculate.

Abdomen pale brown. Tergum 9 in dorsal view with an arcuate anterior incision, and a nearly V-shaped posterior incision, leaving a pair of subtriangular hemitergites, which are obtuse distally and densely haired. Sternum 9 subtrapezoidal, only half the length of tergum 9, almost truncate posteriad (Figure 13H). Ectoproct in dorsal view rectangular, longitudinally elongate (Figure 13G); posterodorsally with a digitiform projections and a pair of posterolaterally directed unguiform projections, posteroventrally with a pair of bifid unguiform projections (Figure 13D,J). Gonocoxite 9 slenderly elongate, with subtriangular proximal half and rodlike distal half, distally extended into a tiny and pointed projection directed medially (Figure 13A,G). Gonocoxite 10 slenderly elongate, longer than gonocoxite 9, proximal half rodlike, angulately incurved, submedially extended into an additional falcate lobe directed posteriad, distally bifurcated, comprising a small, pointed projection directed posterolaterad and a longer, spinous projection directed posteromedially (Figure 13B,H). Fused gonocoxites 11 nearly beam-shaped, convex anteriorly but slightly concaved at middle, laterally connecting to bases of gonocoxites 9 (Figure 13A,G). Hypandrium internum nearly trapezoidal, with lateral margins slightly arcuate (Figure 13B,H).

Female. Body length 5.3 mm; forewing length 12.3 mm, hindwing length 10.8 mm.

Wings with markings much darker than those in male.

Sternum 7 nearly rectangular, with truncate posterior margin (Figure 13L). Segment 8 ventrally without subgenitale. Tergum 9 in lateral view almost as wide as tergum 8 (Figure 13E,K). Bursa copulatrix with colleterial gland tubular and rather short, only anteriorly extended to segment 8 (Figure 13K); basal part of bursa copulatrix nearly round in lateral view, dorsal half with a subtriangular sclerite (Figure 13E,K), which present as a pair of drop-shaped sclerites in ventral view (Figure 13L). Ectoproct small, ovoid (Figure 13E,K).

Materials examined. Holotype ♂, China, Yunnan [Province], Pingbian, Weicheng Umgeb, Mt. Weibaoshan, 25°04′ N 100°20′ E, 2600–2900 m, 10–15/VII/1993, leg. Carolus Holzschuh (CHRR, stored in ETOH). Paratypes 2♀, same data as holotype (CHRR and NMW, stored in ETOH).

Etymology. The specific epithet “*weibaoshanensis*” refers to the type locality of the new species, that is, Mt. Weibaoshan in Yunnan Province, China. It is an adjective, masculine, nominative, singular as an attribute to the genus name.

Distribution. China (Yunnan).

Remarks. The new species should be a member of the *Dilar lijiangensis* species-group based on the slenderly elongate male gonocoxite 9, which is inflated at base, and the male gonocoxite 10 submedially bifurcated. Among the species of the *D. lijiangensis* species-group, the new species appears to be closely related to *Dilar nobilis* Zhang, Liu, H. Aspöck and U. Aspöck, 2015 from China, in having similar characters, for example, the forewing with many dark spots connected with each other, the slenderly elongate male gonocoxite 9 inflated at base, and the slenderly elongate male gonocoxite 10 bifurcated at middle. However, it can be clearly distinguished from the latter species by the gonocoxite 9 with subtriangular proximal half and feebly incurved tip, the bifid male gonocoxite 10 with longer, falcate inner lobe and relatively shorter, bifid outer lobe. In *D. nobilis*, the male gonocoxite 9 is spoon-shaped on proximal half and strongly, anteromedially curved at tip, and the bifid male gonocoxite 10 has a shorter, thin inner lobe and relatively longer, but not bifid outer lobe [26].

#### 3.1.11. *Dilar yucheni* sp. nov.

Figure 2F, Figure 3B–D, Figure 14A–I and 23.

**Figure 14 insects-12-00451-f014:**
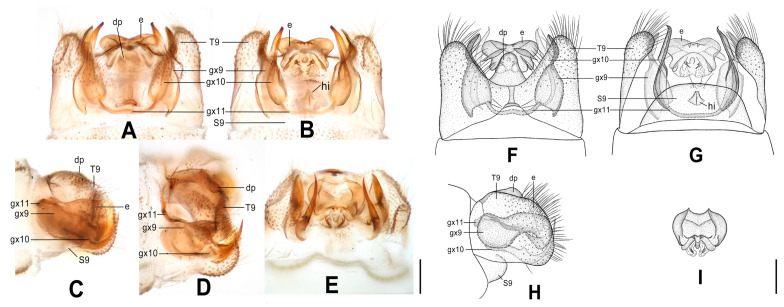
*Dilar yucheni* sp. nov. (**A**–**E**) male holotype, genitalia, photographs: (**A**) dorsal view; (**B**) ventral view; (**C**) lateral view; (**D**) dorsolateral view; (**E**) caudal view. (**F**–**I**) male holotype, genitalia, line drawings: (**F**) dorsal view; (**G**) ventral view; (**H**) lateral view; (**I**) ectoproct, caudal view. dp: dorsoprocessus; e: ectoproct; gx9: gonocoxite 9; gx10: gonocoxite 10; gx11: fused gonocoxites 11; hi: hypandrium internum; S9: sternum 9; T9: tergum 9. Scale bars: 0.2 mm.

Diagnosis. The new species is characterized by the forewing with densely spaced, transverse arcuate stripes and a small immaculate area present distad from median nygma, the presence of weakly sclerotized, subtriangular dorsoprocessus, and the strongly inflated male gonocoxite 9 with semilunar tip slightly curved outward.

Description. Male. Body length 6.5 mm; forewing length 7.8 mm, hindwing length 7.0 mm.

Head generally brown, with light yellow setose tubercles; vertex dark brown. Compound eyes blackish brown. Antenna brown, pedicel distally with dark brown annual stripe, flagellum with medial branches much longer than those branches at both ends, longest branch nearly 4.0 times as long as corresponding flagellomere, distal eight flagellomeres simple.

Prothorax pale brown, pronotum medially with a pair of light yellow ovoid tubercles; meso- and metathorax yellow, mesonotum dark brown on mesoscutellum as well as along anterior and lateral margins, submedially with a pair of dark brown oblique stripes, metanotum paler than mesonotum. Legs generally yellow, but femora dark brown at tip. Wings hyaline, slightly smoky brown. Forewing 2.5 times as long as wide, with densely spaced, transverse arcuate stripes, stripes darker on distal half and costal space, a small immaculate area present distad from median nygma; two nygmata on left forewing (base and middle), three nygmata on right forewing (two at base, one middle); nygmata surrounded by a brownish spot; longitudinal veins pale brown, interrupted by many brown spots; crossveins pale brown. Hindwing 2.5 times as long as wide, much paler than forewing.

Abdomen brown, each pregenital segment dorsally dark brown. Tergum 9 in dorsal view with an arcuate anterior incision, a nearly U-shaped posterior incision, and a dorsoprocessus, leaving a pair of subtriangular hemitergites, which are obtuse distally and densely setose; dorsoprocessus subtriangular, convex posteriorly (Figure 14A,D,F). Sternum 9 nearly trapezoidal, almost half the length of tergum 9 (Figure 14B,G). Ectoproct posterodorsally with a pair of unguiform projections curved posteroventrad, posteroventrally with a digitiform projection and a pair of bifid unguiform projections (Figure 14E,I). Gonocoxite 9 strongly inflated, with distal one third portion slightly curved laterally into a semilunar folded portion, which is much shorter in width to proximal portion of gonocoxite 9 (Figure 14A,D,F). Gonocoxite 10 slenderly elongate, almost as long as gonocoxite 9, with slightly incurved base and slightly outcurved, spinous tip, submedially with a processus connecting to gonocoxite 10 (Figure 14B,G). Fused gonocoxites 11 nearly beam-shaped, slightly anteriorly convex, laterally connecting to bases of gonocoxites 9 (Figure 14A,B,F,G). Hypandrium internum nearly trapezoidal, with lateral margins slightly arcuate (Figure 14B,G).

Female. Unknown.

**Materials examined.** Holotype ♂, China, Xizang Province, Zayü County, Xia Zayü, 1580 m, 9/VIII/2020, Yuchen Zheng (CAU, stored in ETOH).

**Etymology.** The new species is dedicated to Mr. Yuchen Zheng, who collected the holotype of the new species. The specific epithet is a substantive, masculine, genitive, singular of the latinized form of Yuchen as an attribute to the genus name.

**Distribution.** China (Tibet).

**Remarks.** The species is a member of the *D. guangxiensis* species-group due to the presence of dorsoprocessus and the male gonocoxite 10 submedially with a processus connecting to gonocoxite 9. In the *D. guangxiensis* species-group, this new species appears to be closely related to *D. zhangweiae* sp. nov. However, irrespective of the presence of dorsoprocessus on male tergum 9, the new species looks very similar to *D. maculosus* Zhang, Liu, H. Aspöck and U. Aspöck, 2015, from western Yunnan Province, which is not placed in the *D. guangxiensis* species-group. The three species mentioned above have similarities in both external and male genital characters, for example, the presence of dark brown oblique stripes on mesonotum, the similar marking patterns on forewing, the similar modifications of male ectoproct, the strongly inflated male gonocoxite 9 with distal portion curved outward, the male gonocoxite 10 submedially connecting to gonocoxite 9 and spinous at tip. Nonetheless, they can be distinguished by the details of male gonocoxites 9 and 10. In *D. zhangweiae* sp. nov., the distal subtriangular folded portion of male gonocoxite 9 is almost subequal in width to its proximal portion, and the male gonocoxite 10 is bifurcated on distal half. The distal folded portion of male gonocoxite 9 is subtriangular in *D. maculosus* while it is semilunar in *D. yucheni* sp. nov. Both folded portions are much shorter than proximal portion of gonocoxite 9. The male gonocoxite 10 is not bifurcated. Besides, the dorsoprocessus of male tergum 9 is absent in *D. maculosus* [26], but is either a subtrapezoidal sclerite in *D. zhangweiae* sp. nov. or a subtriangular sclerite as in *D. yucheni* sp. nov.

In addition, to clarify the identity of the new species, we carefully compared the holotype of *D.* yucheni sp. nov. and the holotype plus 24 newly collected specimens of *D. maculosus*. To our surprise we found that in some specimens of *D. maculosus* the posteromedial portion of male tergum 9 is weakly sclerotized, thickened or protruding at the location where the dorsoprocessus is generally located (Figure 16A–D). In the holotype of *D. yucheni* sp. nov., the dorsoprocessus is weakly sclerotized and inconspicuous in dorsal view. Thus, the question arises whether the presence of a dorsoprocessus is a consistent feature of all conspecific individuals in certain species of *Dilar*. Here, we consider *D. yucheni* sp. nov. to be a species distinct from *D. maculosus* based on the presence of dorsoprocessus and details of male gonocoxite complexes. Nonetheless, we must be cautious when using the presence/absence of dorsoprocessus to differentiate between species-groups of *Dilar*.

#### 3.1.12. *Dilar zhangweiae* sp. nov.

Figure 2G, Figure 15A–I, and 23.

**Figure 15 insects-12-00451-f015:**
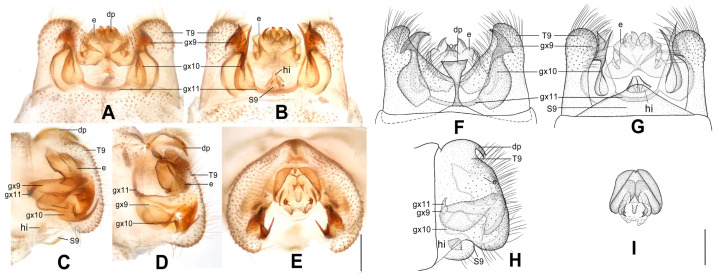
*Dilar zhangweiae* sp. nov. (**A**–**E**) male holotype, genitalia, photographs: (**A**) dorsal view; (**B**) ventral view; (**C**) lateral view; (**D**) dorsolateral view; (**E**) caudal view. (**F**–**I**) male holotype, genitalia, line drawings: (**F**) dorsal view; (**G**) ventral view; (**H**) lateral view; (**I**) ectoproct, caudal view. dp: dorsoprocessus; e: ectoproct; gx9: gonocoxite 9; gx10: gonocoxite 10; gx11: fused gonocoxites 11; hi: hypandrium internum; S9: sternum 9; T9: tergum 9. Scale bars: 0.2 mm.

Diagnosis. The new species is characterized by the forewing with many transverse arcuate stripes and an immaculate area present distad from median nygma, the male tergum 9 with posteriorly widened and truncate dorsoprocessus, the strongly inflated male gonocoxite 9 with subtriangular tip strongly curved outward, and the male gonocoxite 10 bifurcated on distal half. 

Description. Male. Body length 6.0 mm; forewing length 8.9 mm, hindwing length 7.8 mm.

Head generally brown, with light yellow setose tubercles; vertex brown. Compound eyes blackish brown. Antenna generally pale brown, pedicel distally with dark brown annual stripe, flagellum with branches darker than corresponding flagellomere, medial branches much longer than those branches at both ends, longest branch nearly 6.0 times as long as corresponding flagellomere, distal seven flagellomeres simple.

Prothorax yellow, pronotum laterally dark brown, medially with a pair of light yellow ovoid markings; meso- and metathorax yellow, mesonotum dark brown on mesoscutellum as well as along anterior and lateral margins, submedially with a pair of dark brown oblique stripes, metanotum paler than mesonotum, laterally with two pairs of dark brown round markings. Legs generally yellow, but femora and tibiae dark brown at tip. Wings hyaline. Forewing 2.2 times as long as wide, with many transverse arcuate stripes, proximal stripes much darker than remaining distal stripes, an immaculate area present distad from median nygma; two nygmata present, one at base and one middle, both nygmata surrounded by a brownish spot; longitudinal veins light yellow, interrupted by many brown spots; crossveins pale brown. Hindwing 2.2 times as long as wide, much paler than forewing.

Abdomen brown, each pregenital segment dorsally dark brown. Tergum 9 in dorsal view with an arcuate anterior incision, a nearly U-shaped posterior incision, and a dorsoprocessus, leaving a pair of subtriangular hemitergites, which are obtuse distally and densely setose; dorsoprocessus widened and truncate posteriorly, slightly sclerotized on both ends of posterior margin (Figure 15A,D,F). Sternum 9 subtriangular, only half the length of tergum 9 (Figure 15B,G). Ectoproct in dorsal view nearly trapezoidal, with an arcuate anterior incision (Figure 15A,F), posterodorsally with a pair of posterolaterally directed unguiform projections, posteroventrally with a digitiform projection, a pair of bifid unguiform projections and a pair of inflated membranous projections (Figure 15E,I). Gonocoxite 9 strongly inflated, with distal one third strongly curved laterally into a subtriangular folded portion, which is subequal in width to proximal portion of gonocoxite 9 (Figure 15A,F). Gonocoxite 10 almost as long as gonocoxite 9, distal half nearly rectangular, with anterior margin strongly sclerotized and connected to gonocoxite 9, distally bifurcated, comprising a spinous and a broadly subtriangular lobe (Figure 15B,D,G). Fused gonocoxites 11 nearly beam-shaped, laterally connecting to bases of gonocoxites 9 (Figure 15A–D). Hypandrium internum nearly trapezoidal, with lateral margins slightly arcuate (Figure 15B,G).

Female. Unknown.

Materials examined. Holotype ♂, China, Xizang Province, Zayü County, near city, 28°66′ N 97°47′ E, 2362 m, 18/VI/2019, light, Rongrong Shen and Yingnan He (CAU, stored in ETOH). Paratypes 2♂, same data as holotype (CAU, stored in ETOH); 1♂, China, Xizang Province, Zayü County, Xia Zayü, Zayü Farm Laochangbu, 5/VI/2020, light, Xingyue Liu (CAU, stored in ETOH); 1♂, China, Xizang Province, Zayü County, Xia Zayü, Gadui Village, 5/VI/2020, Xingyue Liu (CAU, stored in ETOH).

Etymology. The new species is dedicated to Ms. Wei Zhang, who made considerable contributions on the systematics of *Dilar* from East and Southeast Asia in the past few years. The specific epithet is a substantive, feminine, genitive, singular of the latinized form of Zhang Wei as an attribute to the genus name.

Distribution. China (Tibet).

Remarks. The new species is a member of the *D. guangxiensis* species-group by the presence of dorsoprocessus on male tergum 9 and the male gonocoxite 10 submedially connecting to gonocoxite 9. For more differentiation, see Remarks under *D. yucheni* sp. nov.

**Figure 16 insects-12-00451-f016:**
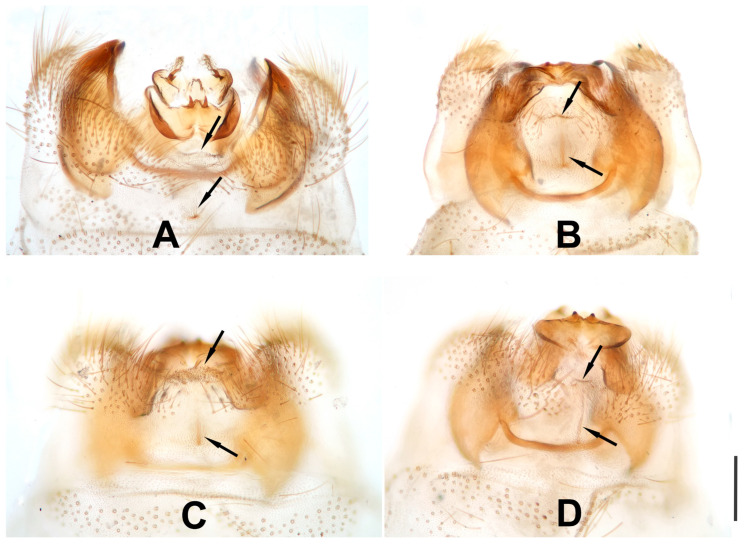
*Dilar maculosus* Zhang, Liu, H. Aspöck and U. Aspöck, 2015, male tergum 9, ventral view, photographs. (**A**) holotype, China: Yunnan, Baoshan, Nankang; (**B**–**D**) China, Yunnan, Baoshan, Nankang, Gaoligong Natural Park. Arrow indicating the weakly sclerotized and protruded at posteromedially of tergum 9. Scale bars: 0.2 mm.

**Figure 17 insects-12-00451-f017:**
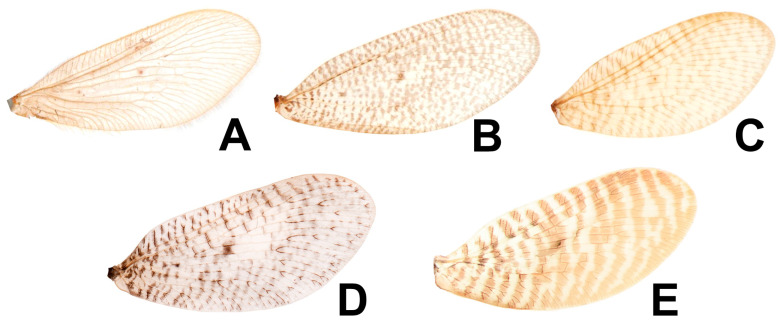
Five marking patterns of forewing. (**A**) type (i): wing sparsely spotted or almost immaculate; (**B**) type (ii): wing with dense and scattered spots throughout; (**C**) type (iii): wing with many short, arcuate stripes throughout; (**D**) type (vi): wing intensively spotted on base and distal 1/3, medially with a distinct immaculate area, median nygma generally surrounded by a large and dark spot; (**E**) type (v): wing with a series of long, transverse bands.

### 3.2. Biogeography

The richness of species diversity and global distribution pattern of *Dilar* are presented below based on comprehensive data (Figure 18, Figure 19, Figure 20, Figure 21, Figure 22, Figure 23, Figure 24, Figure 25 and Figure 26). The distribution range of *Dilar* comprises the southern parts of the Palearctic region and almost all major parts of the Oriental region, and stretches across a wide elevation range from 0 to 3900 m. The correlation between elevation and latitude for the different regions was shown in Figure 27. The results suggest that elevation and latitude are only significantly correlated in two regions, that is, Central Asia and southern part of South Asia. Eight areas of endemism were distinguished. The following is a brief description of each area’s faunal components.

Europe, North Africa and West Asia (Region 1). The 13 species from this area account for 16% of the global fauna of *Dilar* (Figure 20). In this region, adults can be found from March to October across a wide range of elevation (0–2500 m), but particularly in summer from May to August. Six species are restricted to the Iberian Peninsula, which has the richest species diversity in Europe. Only a single species is known from North Africa, namely *Dilar bolivari* Navás, 1903. Three species are restricted to southern France and the Apennine Peninsula, while another three species are distributed in Eastern Europe and West Asia. The species of the Iberian Peninsula usually have sympatric distributions, for example, *Dilar meridionalis* Hagen, 1866, occurs sympatrically with (i) *Dilar dissimilis* Navás, 1903, in northeastern Spain (Pine de Ebro, Retuerta de Pina), (ii) *Dilar saldubensis* Navás, 1902, in northern Portugal (Ria Sabor and Serra do Marao), (iii) *Dilar nevadensis* Rambur, 1838, and (iv) *Dilar juniperi* Monserrat, 1988, in southern Spain (Sierra Nevada and Collado de los Jardines). In West Asia, *Dilar fuscus* U. Aspöck, Liu and H. Aspöck, 2015, and *Dilar turcicus* Hagen, 1858, are sympatric in parts of western Turkey (Aydin). Interestingly, the three species from the Balkan Peninsula and Anatolia appear to be only distantly related to others from Europe and Africa but closely related to some from East Asia on account of similarities between the forewing patterns (wing entirely brown without distinct markings in *D. fuscus*), dorsoprocessus (elongated rod-like in *D. turcicus*) and the complex of male gonocoxites 9, 10 and 11 (gonocoxite 9 short, shield-like and gonocoxite 10 slenderly elongate in *Dilar anatolicus* U. Aspöck, Liu and H. Aspöck, 2015 [28]).

Central Asia (Region 2). The five species from this area account for 6% of global fauna. In this region, adults occur from April to August at an elevation range of 950–3200 m. It is notable that the elevation and latitude are significantly as well as negatively correlated in this region (*R*^2^ = 0.3242, *p* < 0.01). Thus, a decrease in elevation was accompanied by an increase in latitude with regard to the species’ distribution localities (Figure 27B). All five species are restricted to this region (Figure 21). *Dilar kirgisus* H. Aspöck and U. Aspöck, 1967, is herein newly recorded from western Xinjiang (China), an area which is inferred to have affinity with the fauna of Central Asia. *Dilar kirgisus* appears to have a much broader distribution than other Central Asian species. *Dilar hornei* McLachlan, 1869, which was originally recorded from northwestern India [30,35,36], was reported from Kurseong, West Bengal, northeastern India by Ghosh [37]. However, considering the distinct faunal differences between northwestern and northeastern India, this record is probably based on a misidentified specimen and was excluded from distribution data for *D. hornei*.

**Figure 27 insects-12-00451-f027:**
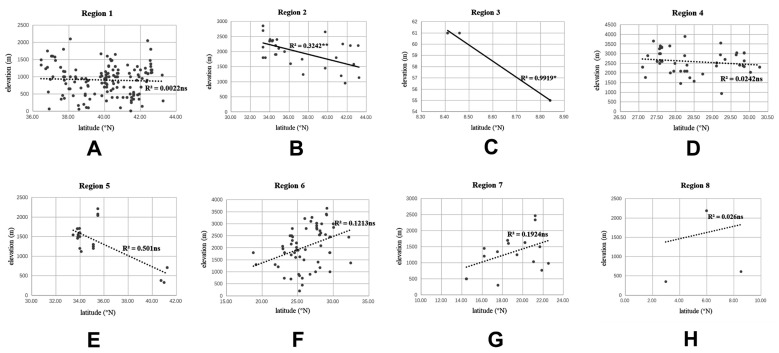
Correlation between elevation and latitude in each region, (**A**) Europe, North Africa and West Asia (Region 1); (**B**) Central Asia (Region 2); (**C**) south part of South Asia (Region 3); (**D**) southern Tibet and adjacent area (Region 4); (**E**) Palearctic part of East Asia (Region 5); (**F**) Oriental China (Region 6); (**G**) Indochina Peninsula (Region 7); (**H**) Indo-Malaysia (Region 8). Significance levels: **, *p* < 0.01; *, *p* < 0.05. ns = not significant.

Southern part of South Asia (Region 3). The six species from this area account for 7% of global fauna (Figure 22). In this region, *Dilar* species inhabit an extremely low and narrow altitude range of 55–61 m and occur in early spring (February to March) or autumn (September to October). Based on our analysis, the latitude of the distribution locality is significantly correlated with altitude (*R*^2^ = 0.9919, *p* < 0.05). However, the results must await further verification since only three sets of data were currently available. Three species, that is, *Dilar clavatus* Li, U. Aspöck, H. Aspöck and Liu, 2020, *Dilar miralobatus* Li, U. Aspöck, H. Aspöck and Liu, 2020, and *D. truncatus*, share a sympatric distribution in northern Sri Lanka. Species from Sri Lanka and from central and southern India have generally similar characters related to the complex of male gonocoxites 9, 10 and 11, indicating a close relationship among the species in Region 3.

Southern Tibet and adjacent area (Region 4). The five species from this region account for 6% of global fauna (Figure 23). Adults occur from April to August in the elevation range of 939–3900 m. The distribution of all five species is not only concentrated, but also restricted, to the Himalayas. The species are assigned to the *D. guangxiensis* species-group [26], which further includes species from southern China, indicating a close relationship between those from the Himalayas and southern China.

Palearctic part of East Asia (Region 5). The seven species from this area account for 8% of global fauna (Figure 24). In this region, adults occur from May to August, in the elevation range of 330–2213 m. *Dilar septentrionalis* Navás, 1912, is widely distributed from northern China and adjacent areas such as Korea and the Russian Far East. *Dilar hastatus* Zhang, Liu, H. Aspöck and U. Aspöck, 2014, and *Dilar sinicus* Nakahara, 1957, are restricted to northern China. *Dilar hikosanus* Nakahara, 1955, and *Dilar japonicus* McLachlan, 1883, are restricted to Japan. Additionally, one species, *Dilar spectabilis* Zhang, Liu, H. Aspöck and U. Aspöck, 2014, is distributed in both northern and southern China. Sympatric distribution is found in *D. sinicus*, *D. spectabilis* and *Dilar taibaishanus* Zhang, Liu, H. Aspöck and U. Aspöck, 2014, in Haoping Temple, Shaanxi.

Oriental China (Region 6). The 30 species distributed in this area account for 36% of global fauna (Figure 25). In this region, *Dilar* species inhabit a wide elevation range from 200 to 3800 m, and the adults mainly occur from May to August. A total of 23 species inhabit the southern part of mainland China, and four species are restricted to the islands of Taiwan and Hainan. Only three species, that is, *D. spectabilis*, *Dilar cornutus* Zhang, Liu, H. Aspöck and U. Aspöck, 2015, and *Dilar nobilis* Zhang, Liu, H. Aspöck and U. Aspöck, 2015, have distributions beyond the area. Sympatric distributions are found between (i) *Dilar guangxiensis* Zhang, Liu, H. Aspöck and U. Aspöck, 2015, and *D. longidens* Zhang, Liu, H. Aspöck and U. Aspöck, 2015, from Huaping Natural Reserve, Guangxi; (ii) *D. cornutus* and *D. maculosus* in Nankang and Baihualing, Yunnan; (iii) *D. lijiangensis* and *Dilar montanus* Yang, 1992, from Gaoshan Botanical Garden, Yunnan; and (iv) *Dilar pallidus* Nakahara, 1955, and *Dilar taiwanensis* Banks, 1937, from Nantou, Taiwan. To date, 13 species are recorded from Yunnan and 10 of which are endemic. Nevertheless, the *Dilar* fauna from Yunnan appears to have a complex evolution and was formed by multiple sources since a few species, for example, *D. cornutus*, *D. nobilis*, *Dilar dulongjiangensis* Zhang, Liu, H. Aspöck and U. Aspöck, 2015, and *D. maculosus*, also occur in Indochina or are closely related to species from Indochina and Tibet.

Indochina Peninsula (Region 7). With six newly described species, the species number of *Dilar* from this area has increased to 16, accounting for 19% of global fauna (Figure 26). In this region, adults mainly occur from March to June in an elevation range of 295–2470 m. Most species are restricted to this area except *D. cornutus*, *D. nobilis*, and *Dilar marmoratus* (Banks, 1931). The former two species are also distributed in southern China, while the latter is also known from the Malay Peninsula. *Dilar laoticus* sp. nov. and *D. forcipatus* sp. nov. are sympatric in Ban Saleui, Houaphanh, northeastern Laos, while *Dilar malickyi* sp. nov. and *D. marmoratus* share a sympatric distribution in Tham Than Lod, northern Thailand. Since *D. aspoeckorum* and *D. laoticus* sp. nov. are assigned to the *D. guangxiensis* species-group, a close faunal relationship between Indochina and southern China is implied.

Indo-Malaysia (Region 8). Only four species are known from this area, accounting for 5% of global fauna (Figure 26). In this region, adults mainly occur from March to June and in November, in an elevation range of 350–2190 m. Three species are restricted to this area, while *D. marmoratus* has also been recorded from Indochina. Based on male genital characters, *Dilar grandis* (Banks, 1931) and *D. marmoratus* are assigned to the *D. guangxiensis* species-group and the *Dilar hastatus* species-group, respectively [27], indicating a possibly ancient faunal connection between this region and East Asia. 

## 4. Discussion

### 4.1. Morphological Diversity

Together with the presently described 12 new species, a total of 87 extant species of *Dilar* exist worldwide. Zhang et al. [26] proposed five species-groups of *Dilar* in the Oriental region based on male genital characters. Following this subdivision, only 36 species could be assigned to the species-groups, including four of the new species herein described, while the remaining 51 species are difficult to place in any species-group [6,26,27,29].

The patterns of wing markings of *Dilar* differ among species, despite a striking variability in size, shape and coloration that has been observed among conspecific individuals in five species, that is, *Dilar geometroides* Aspöck and Aspöck, 1968, *D. harmandi*, *Dilar muliensis* Li and Liu, 2020, *D. taiwanensis* and *D. turcicus* Hagen, 1858 [6,24,25,28]. Preliminarily, the forewing marking patterns are divided into five types: (i) wing sparsely spotted or almost immaculate (Figure 17A); (ii) wing with dense and scattered spots throughout (Figure 17B); (iii) wing with many short, arcuate stripes throughout (Figure 17C); (iv) wing intensively speckled on base and distal 1/3, medially with a distinct immaculate area, median nygma generally surrounded by a large and dark spot (Figure 17D); and (v) wing with a series of long, transverse bands (Figure 17E). Of these, types (ii) and (iii) might be plesiomorphic or have evolved convergently on multiple occasions since the two patterns are present in almost all *Dilar* species-groups as well as all species of the seven above-mentioned regions. In contrast, types (i), (iv) and (v) are present in only few species found either in western Asia, southern mainland China, Indochina Peninsula or Indo-Malaysia. These types of patterns are likely to be apomorphic and of phylogenetic relevance. Among the 12 new species herein described, type (i) is present in *D. malickyi* sp. nov., type (iii) in *D. rauschorum* sp. nov., and type (v) in both *D. forcipatus* sp. nov. and *D. striatus* sp. nov. The wing patterns of the remaining eight species all belong to type (iv).

As concerns the male genital sclerites, some unique or notable characters were discovered. For example, we found a pentagonal dorsoprocessus on tergum 9, which is unique to *D. rauschorum* sp. nov. In other species of *Dilar* that possess a dorsoprocessus, the structure is generally a subtriangular, rectangular, or trapezoidal sclerite. Regarding the ectoproct of the two new species, that is, *D. forcipatus* sp. nov. and *D. nujianganus* sp. nov., the former has a digitiform projection and the latter a deeply bifurcated medial projection on the ectoproct, neither character has been previously described in other species of *Dilar*. The gonocoxite 10 in these two new species, as well as in *D. rotundatus*, is inflated on the distal half, otherwise this sclerite is generally narrowed on the distal half and tapered at the tip. The fused gonocoxites 11 are bifurcated on both ends or form a pair of longitudinal projections in *D. malickyi* sp. nov. and *D. forcipatus* sp. nov., although this sclerite is not a well-modified beam-shaped structure as in most species of *Dilar*. The male genital characters are complex throughout the species; thus, they serve to support the division into species-groups and may bear phylogenetic inference on the interspecific relationships in *Dilar*, which, however, will be comprehensively compared and evaluated in the near future.

### 4.2. Biogeography

Concerning Neuroptera as a whole, the Palearctic region has the highest diversity with 1440 recorded species, followed by the Afrotropical region (1390 species), the Oriental region (1260 species), the Neotropical region (1170 species), the Austroceania region (840 species) and the Nearctic region (480 species) [4]. However, the species diversity of *Dilar* is richest in the Oriental region, with 55 species. Furthermore, almost all of these 55 species are endemic to the Oriental region except *D. spectabilis* which is also recorded from the Palearctic region. The remaining 31 species of *Dilar* are restricted to the Palearctic region. It is notable that the proportion of endemic species of each of the eight areas of endemism is very high, and is ranked as follows: Europe, North Africa and West Asia (100%), Central Asia (100%), southern part of South Asia (100%), southern Tibet and adjacent area (100%), Oriental China (90%), Palearctic part of East Asia (86%), Indochina Peninsula (81%), Indo-Malaysia (75%). The phenomenon of high endemism may be due to the weak dispersal capacity of *Dilar*. Sympatric distribution is another common phenomenon in *Dilar* since frequently two or three species occur at the same locality. The highly disparate morphologies in male genitalia may account for effective reproductive isolation among sympatric congeneric species. Joint glacial refugial centers in the sense of de Lattin [38] may be another contributing factor to this phenomenon.

The species-rich fauna of Oriental China appears to be composed of systematically heterogeneous species since representatives of all five-known species-groups of *Dilar* occur in this region [6,26]. Despite high endemism, Oriental China has certain faunal connections with adjacent regions, that is, Palearctic part of East Asia, Indochina, Indo-Malaysia, and southern Tibet by sharing several common species or similar species of the same species-group. Within the provincial divisions of 40 countries where species of *Dilar* occur, Yunnan possesses the highest number of species (13 species). Since Yunnan has vast mountainous areas and an unusually rich spectrum of vegetation types, numerous geographical and ecological isolations were likely produced, thus accelerating the speciation of *Dilar* in the region.

It is clear that more than 75% species of *Dilar* are able to inhabit high altitudes of mountains (>1100 m). Some of the newly described dilarid species in the present paper represent such examples, for example, *D. daweishanensis* sp. nov., from an altitude of 1962 m, *D. cangyuanensis* sp. nov., from 1800 m, *D. nujianganus* sp. nov., from 2775 m, and *D. weibaoshanensis* sp. nov., from 2600–2900 m, all from Yunnan, in addition, *D. yucheni* sp. nov., at an altitude of 1580 m, and *D. zhangweiae* sp. nov. at an altitude of 2360 m from Tibet. The phenomenon of living at high altitudes has been discussed and dealt with in different approaches in De Lattin’s [38] chapter on the Oriental region. More recently, the interlocking of Oriental and Palearctic distribution areas was analyzed within the context of Odonata [39] and Raphidioptera (Inocelliidae) [40,41]. The main argument and justification for this interpretation is the fact of their present-day occurrence at high altitudes. Over the course of postglacial warmings/rising temperatures, the insects were presumably only able to escape by ascending up the mountains. There are exceptions, however. Some species inhabit distinctly lower elevations, for example, *D. malickyi* sp. nov., known from 500 m, and *D. phraenus* sp. nov., occurring at 295 m, both from Thailand. In addition, three species from Sri Lanka, for example, *D. clavatus*, *D. miralobatus* and *D. truncatus* are known only from 55–61 m. Notably, in our analysis of the distribution data, the correlation between latitude and elevation was not significant in most of the aforementioned regions in which species of *Dilar* occur (Figure 27). These exceptions refer to Central Asia (Region 2) and southern part of South Asia (Region 3), in which the increase of latitude is significantly correlated to the decrease of elevation. However, it must be cautioned that the data analyzed for Region 3 was not entirely sufficient. A similar trend is apparent in the Palearctic part of East Asia (Region 5), although statistically the correlation was not significant. Concerning the Oriental China (Region 6), Indochina (Region 7) and Indo-Malaysia (Region 8), and despite insignificant correlations, species from higher latitude localities tended to inhabit higher elevation areas. Therefore, it is apparent that the pattern of distributional preference toward elevation in *Dilar* is complex. Some Palearctic elements of *Dilar* at the transition zone between the Palearctic and Oriental realms may be expected to occur in high elevation areas due to postglacial rising temperature. Conversely, the typical Oriental elements of *Dilar* may be unable to conform to this pattern since they inhabit areas with a broad spectrum of elevations. As the most species-rich and widespread genus of Dilaridae, the strong adaptability of *Dilar* to heterogeneous habitats could be an alternative explanation.

## Figures and Tables

**Figure 18 insects-12-00451-f018:**
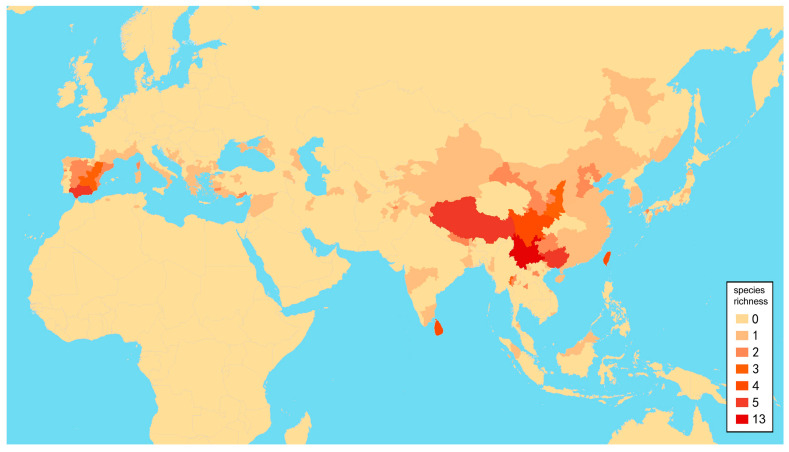
Pattern of species richness of the genus *Dilar*.

**Figure 19 insects-12-00451-f019:**
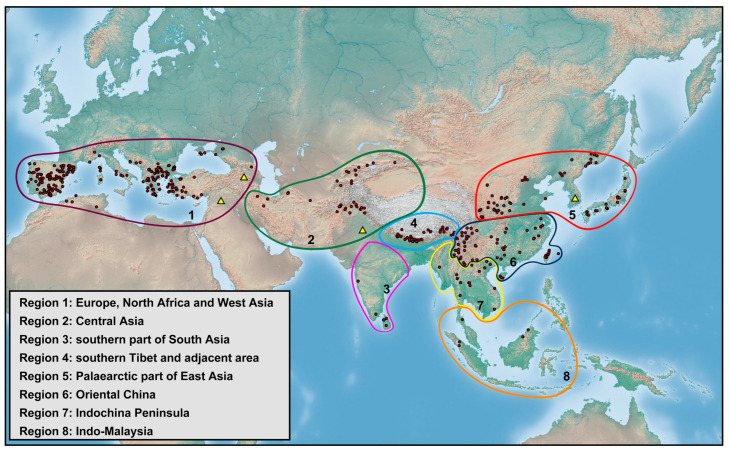
Distribution map of the species of *Dilar* worldwide. Triangles indicate specimens with imprecise locality.

**Figure 20 insects-12-00451-f020:**
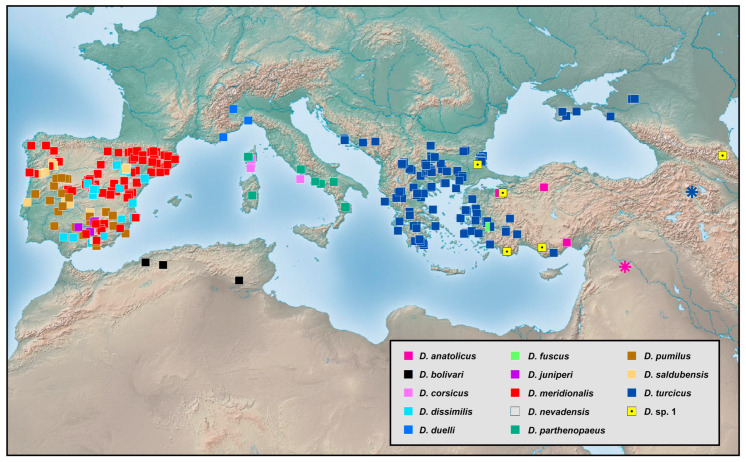
Distribution map of the species of *Dilar* from Europe, North Africa and West Asia. (Region 1 in Figure 19) Asterisk indicates specimen with imprecise locality.

**Figure 21 insects-12-00451-f021:**
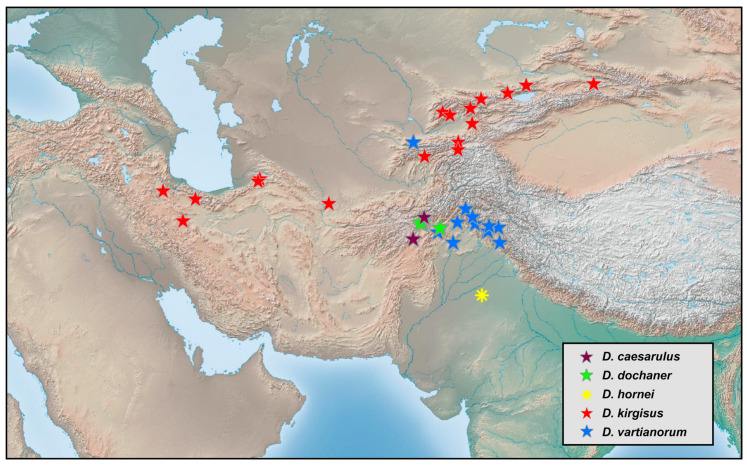
Distribution map of the species of *Dilar* from Central Asia (Region 2 in Figure 19).

**Figure 22 insects-12-00451-f022:**
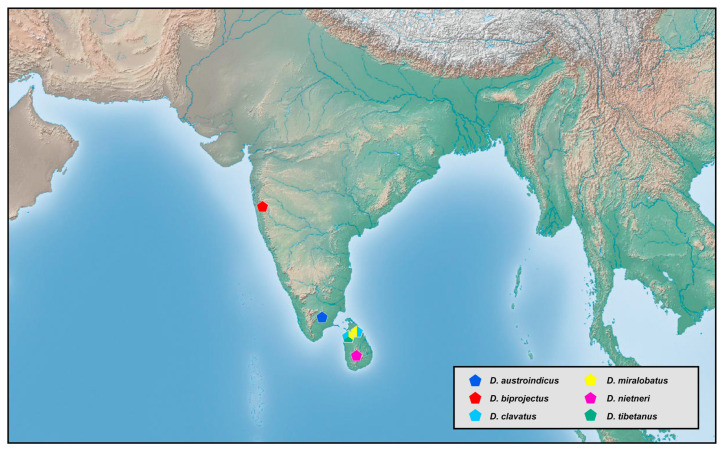
Distribution map of the species of *Dilar* from the southern part of South Asia (Region 3 in Figure 19).

**Figure 23 insects-12-00451-f023:**
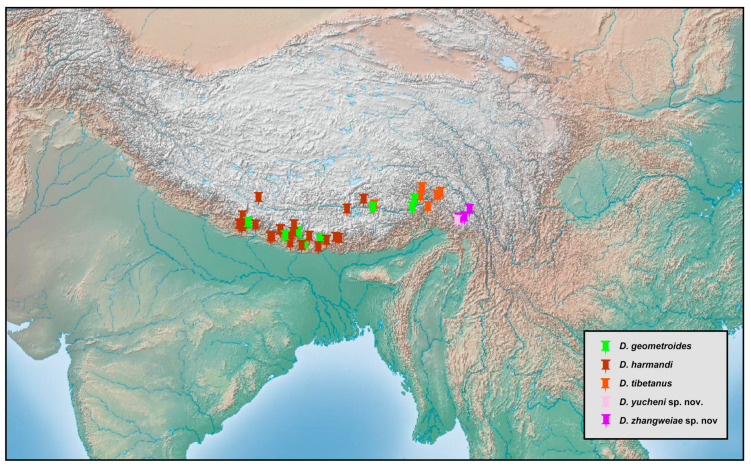
Distribution map of the species of *Dilar* from southern Tibet and adjacent area (Region 4 in Figure 19).

**Figure 24 insects-12-00451-f024:**
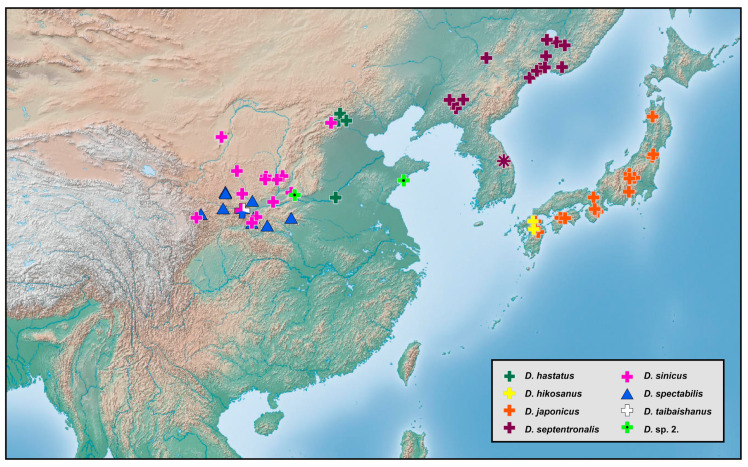
Distribution map of the species of *Dilar* from the Palearctic part of East Asia (Region 5 in Figure 19). Asterisk indicates specimen with imprecise locality.

**Figure 25 insects-12-00451-f025:**
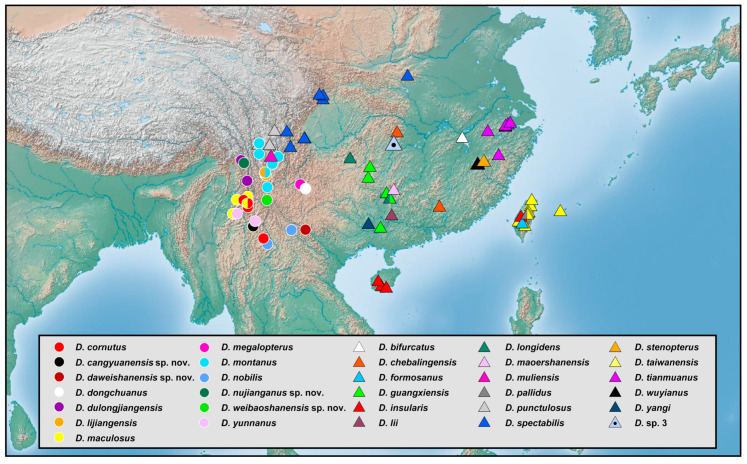
Distribution map of the species of *Dilar* from Oriental China (Region 6 in Figure 19).

**Figure 26 insects-12-00451-f026:**
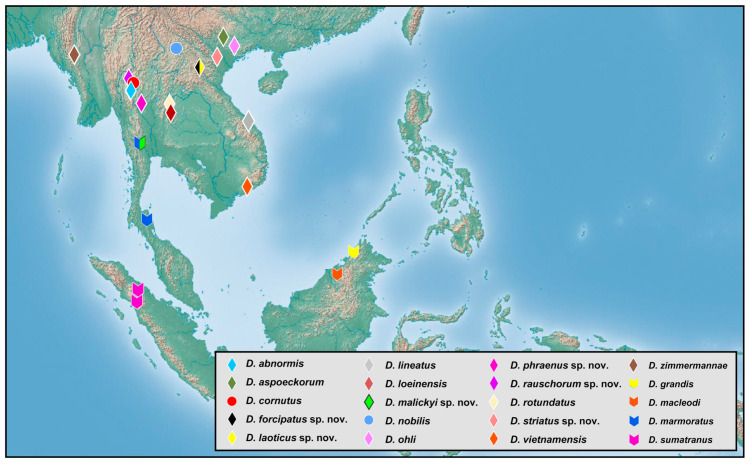
Distribution map of the species of *Dilar* from Indochina Peninsula (Region 7 in Figure 19) and Indo-Malaysia (Region 8 in Figure 19).

## Data Availability

All data is available in the present paper and Appendix A.

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
