# Peer review of "Mining the Species Diversity of Lacewings: New Species of the Pleasing Lacewing Genus Dilar Rambur, 1838 (Neuroptera, Dilaridae) from the Oriental Region†"

_insects, 2021, doi:10.3390/insects12050451_

Round 1

Reviewer 1 Report

I would say that this manuscript should be accepted for the publication of Insects after some minor revisions. This manuscript is an important entomological paper because the results contain many thought-provoking information of new species at rare lacewing groups.

Minor points

1: “little known” in the title (L2) should be deleted because you too much emphasize on the “little known” or same meaning words, though the text. I would feel that they are a bit tenacious. This expansion is at least not suitable for the title of scientific paper.

2: You include some Chinese characters in the text (e.g. L336, 905, 992, 994, 996). I would understand that you use Chinese characters for better understanding of place names. However, many non-Chinese potential readers cannot understand them. I would recommend you that you should delete these Chinese characters.

3: Conclusions (L1273 to 1281) should be deleted because this manuscript is very long and they are largely overlapped Summary.

Author Response

Dear reviewer,

We gratefully thank you for your time spending making your constructive suggestions on our manuscript. Each suggested comment was accurately incorporated and considered. Here are our responses point by point to the comments.

Point 1:“little known” in the title (L2) should be deleted because you too much emphasize on the “little known” or same meaning words, though the text. I would feel that they are a bit tenacious. This expansion is at least not suitable for the title of scientific paper. 

Response 1: We appreciate for your valuable comment. We would like to remove “little known” in the title as you suggested.

Point 2: You include some Chinese characters in the text (e.g. L336, 905, 992, 994, 996). I would understand that you use Chinese characters for better understanding of place names. However, many non-Chinese potential readers cannot understand them. I would recommend you that you should delete these Chinese characters.

Response 2: Thank you for pointing this out. We have removed the Chinese characters in the whole manuscript.

Point 3: Conclusions (L1273 to 1281) should be deleted because this manuscript is very long and they are largely overlapped Summary.

Response 3: Thank you for your constructive suggestion. We agree to delete the Chapter Conclusion (L1273 to 1281) in the manuscript.

Thank you,

Authors

Reviewer 2 Report

Dear Authors,

The information presented in this manuscript is important to understand the species diversity of lacewings. Findings are novel and important for further understanding of little known lacewing species in the Order Neuroptera. The manuscript is well written with a lot of new taxonomic information about the lacewing Genus Dilar Rambur, 1838 in the family Dilaridae.

I recommend this manuscript to be published in Insect.

Thank you,

Reviewer

 Line 1249: the reference 41 is missing in the reference list.

Line 1295: Remove the extra letter D

Author Response

Dear reviewer,

We gratefully thank you for your careful read and comments on our manuscript. We have carefully taken your comments in preparing our revision. The following summarizes our responses point by point to your comments.

Point 1: the reference 41 is missing in the reference list.

Response 1: Much appreciation to your careful review. We sincerely apologized for our omission of the number of 41 in the chapter Reference. The corresponding correction was made in the revised manuscript.

Point 2: Remove the extra letter D.

Response 2: Thank you for pointing this out. Because DDr. John Plant has two doctorates, thus we hope to keep this way of writing.

Thank you,

Authors